# CHART-RVR: REINFORCEMENT LEARNING WITH VERIFIABLE REWARDS FOR EXPLAINABLE CHART REASONING

## ABSTRACT

The capabilities of Large Vision-Language Models (LVLMs) have reached state-of-the-art on many visual reasoning tasks, including chart reasoning, yet they still falter on out-of-distribution (OOD) data, and degrade further when asked to produce their chain-of-thought (CoT) rationales, limiting explainability. We present **Chart-RVR**, a general framework that fine-tunes LVLMs to be more *robust* and *explainable* for chart reasoning by coupling Group Relative Policy Optimization (GRPO) with automatically verifiable rewards. Our framework comprises of three rewards that maximize: (i) correct chart-type classification, (ii) faithful chart table reconstruction, and (iii) process conformity. Applied to 3-billion-parameter LVLMs, Chart-RVR consistently outperforms standard supervised fine-tuning (SFT) on both in-distribution and out-of-distribution datasets, closing the OOD performance gap while improving rationale fidelity. The resulting models, the **Chart-RVR-3B** series, achieve state-of-the-art results on six chart-reasoning benchmarks spanning in-domain and OOD settings, surpassing all existing models of comparable size. Beyond accuracy, Chart-RVR yields more interpretable CoT rationales, strengthening trust and reliability - showcasing the power of verifiable rewards with GRPO for training reliable, interpretable chart-reasoning models. The code can be found at `https://anonymous.4open.science/r/chart-rvr-730D` for reproducibility.

## 1 INTRODUCTION

Charts are a cornerstone of visual communication and are widely used in finance, healthcare, public policy, and beyond. Experts and non-experts alike rely on them to make judgments that shape policy, allocate resources, and drive strategic investments. Automating the interpretation of such figures is, therefore, a high-value AI problem. Unlike natural images, often described by high-level semantics (e.g., "a dog on a table"), charts encode information through *precise* spatial and numerically aligned relationships. Chart reasoning is inherently *entangled*: structured data representations are tightly interwoven with visual design choices. Consequently, any chart-reasoning model must disentangle these components during decision making. Large Vision–Language Models (LVLMs), pre-trained on billions of image–text pairs, have demonstrated good performance on general visual question-answering benchmarks, including chart reasoning.

Although pre-trained LVLMs are successful on chart reasoning benchmarks, recent studies (Islam et al., 2024) reveal two systematic weaknesses. First, even when an LVLM answers in-distribution (ID) chart questions correctly, its performance significantly collapses on out-of-distribution (OOD) datasets that differ only in visual style, color palette, etc. Second, and more importantly, attempts to elicit model rationales via chain-of-thought (CoT) prompting not only fail to improve accuracy but often *harm* it (Zhang et al., 2025a; Turpin et al., 2023), generating incoherent or hallucinated reasoning traces. This brittleness undermines trust, as a system that cannot explain the reasoning process is unlikely ever to be widely adopted by stakeholders, no matter how impressive its predictive accuracy. This problem is particularly severe in relatively smaller LVLMs (2–3 billion parameters), which are more likely to be used in edge devices and for efficient chart understanding.

To alleviate these problems, current state-of-the-art approaches (Masry et al., 2025b; Carbune et al., 2024; Zhang et al., 2024) utilize supervised fine-tuning (SFT) with labeled chart datasets consisting of questions and step-by-step answer computation traces to improve LVLMs on chart reasoning. However, these methods have moderate success in generalizing to OOD data, and most methods underperform their untrained counterparts, implying that SFT decreases generalizability in chart LVLMs. This behavior is a direct consequence of SFT's goal, maximizing the likelihood of human demonstrations, which incentivizes token-level imitation of reasoning traces rather than verifiable task success.

To alleviate the shortcomings of SFT, which trains models to imitate labeled examples and thus inherits dataset-specific styles and biases, a parallel line of new research utilizes Reinforcement Fine-Tuning (RFT) to fine-tune LVLMs (Liu et al., 2025b). RFT optimizes outcomes by (i) prompting the model and sampling candidate responses, (ii) evaluating feedback (from human preferences or verifiable functions), and (iii) updating the model toward higher-scoring behavior while staying close to the reference (starting untrained) model. Multiple studies have demonstrated improved reasoning (Guo et al., 2025; Shao et al., 2024) and generalization (Chu et al., 2025) abilities of RFT compared to SFT. A common form of RFT is Direct Preference Optimization (DPO), which aligns model generations with human preferences (Xie et al., 2024; Zhang et al., 2025b). However, collecting high-quality preferences for thousands of multi-step numeric explanations is prohibitively expensive, while synthetic chart data does not effectively capture visual diversity. As a solution, recent research has introduced Group Relative Policy Optimization (GRPO) as a lightweight objective that ranks multiple sampled responses for the same prompt and updates the policy toward those with higher rewards relative to the group. By optimizing relative quality between candidates rather than imitating traces, GRPO provides stable training for large vision–language models. More recently, *verifiable rewards* - automatic checks that score outputs on an objective criterion (e.g., format adherence and success on intermediate subgoals) have been successfully utilized to fine-tune LVLMs. Verifiable rewards are machine-checkable, deterministic signals that plug naturally into GRPO, yielding dense, low-variance feedback across multiple candidates and aligning the model with what can be *verified* rather than merely *imitated*.

Building on this premise, we present **Chart-RVR**, a general-purpose reinforcement learning framework that combines GRPO with verifiable rewards tailored to chart reasoning. Chart-RVR utilizes verifiable surrogate task rewards that score a policy's performance on chart type prediction and chart table reconstruction, followed by a verifiable process conformity reward, which incentivizes the model's reasoning process to stylistically follow an algorithmic skeleton, improving robustness under format/domain shift and producing logically coherent CoT rationales. We demonstrate that models trained with the Chart-RVR framework achieve state-of-the-art prediction performance on 6 diverse chart benchmarks and also provide more explainable rationales, improving interpretability. More specifically, our contributions are as follows:

- We propose Chart-RVR, the first general-purpose reinforcement learning framework with verifiable surrogate-task rewards: chart type prediction and chart table reconstruction for improved chart reasoning.
- We present the Chart-RVR-3B series of models, the best state-of-the-art chart-reasoning models of their size (2–3 billion parameters) trained using Chart-RVR. Our method achieves benchmark performance on 6 diverse chart-reasoning benchmarks, including OOD settings.
- We also empirically demonstrate that Chart-RVR produces benchmark results on the surrogate tasks and generates explainable chain-of-thought rationales.

## 2 RELATED WORK

**Chart Reasoning.** Chart reasoning has been an active area of research recently. Benchmarks for studying chart-related downstream tasks, such as chart-to-table conversion, chart captioning, chart factoid-based question answering, etc., have been widely utilized to evaluate vision language models. Multiple chart-specific VLMs have been proposed, such as Unichart (Masry et al., 2023), MatCha (Liu et al., 2023), Pix2Struct (Lee et al., 2023), etc., with considerable success in some of the downstream tasks. However, most of the proposed models struggle when the complexity of the questions increases, which requires relatively deeper reasoning. In addition, with deeper reasoning, it is also imperative to output explanations with the final answers to improve trust in the

models. Some new benchmarks, such as Hegde et al. (2025); Ma et al. (2025), have been proposed to measure both reasoning and accuracy performance in tandem. As a consequence, newer chart reasoning models such as Chartgemma (Masry et al., 2025b), TinyChart (Zhang et al., 2024) have been proposed to output rationales with their predictions. Some other works like ChartAssistant (Meng et al., 2024), ChartBench (Xu et al., 2023), ChartInsights (Wu et al., 2024), etc., have been proposed with newer and more challenging datasets. Newer approaches utilize contrastive learning and graph-based methods to improve CoT performance on charts (Dai et al., 2025) or use visual tools to focus on chart images to answer conflicting questions (Fu et al., 2025).

**Chain-of-thought in LVLMs.** Chain-of-thought entails prompting LLMs to think step by step, before outputting the final prediction, and provides improvement in both performance and interpretability of LLMs through explicit natural language reasoning traces (Wei et al., 2022). However, similar observations are not observed in LVLMs, where CoT-type prompting significantly degrades performance (Zhang et al., 2025a), especially in smaller models. Several approaches have attempted to improve CoT in LVLMs with further pre-training (Xu et al., 2024), Reinforcement Learning (Zhang et al., 2025a; Xie et al., 2024; Liu et al., 2025b), etc. The degraded CoT performance also hinders the explainability of LVLMs (Jiaqi et al., 2025).

Parallel to our work, BigCharts-R1 Masry et al. proposes a larger and more diverse chart reasoning dataset, incorporating existing chart images along with images crawled from the web. It uses GRPO to train the Qwen family of LVLMs on the BigCharts dataset. In contrast, our work is a methodological recipe for fine-tuning small LVLMs with verifiable rewards. It is architecture agnostic and validated on Qwen-VL, Gemma-3, and InternVL-3.5 family of LVLMs. Our motivation is to provide a strong and generalizable fine-tuning methodology that can utilize well-benchmarked/validated datasets (e.g. ChartQA, ChartFC, PlotQA) for LVLM fine-tuning.

## 3 METHODOLOGY

### 3.1 PROBLEM SETUP AND NOTATION

Let $\mathcal{D} = \{(x_i, q_i, y_i^*, a_i^*)\}_{i=1}^{N}$ be a dataset of chart images $x_i \in \mathcal{X}$, natural-language queries $q_i \in \mathcal{Q}$, ground-truth answers $y_i^* \in \mathcal{Y}$, and expert rationales $a_i^* \in \mathcal{A}$. Each rationale decomposes into three components:

$$a_i^* = (c_i^*, T_i^*, w_i^*),$$

where $c_i^* \in \mathcal{C}$ is the chart type, $T_i^* \in \mathcal{T}$ is the underlying table representation, and $w_i^* \in \mathcal{W}$ is a natural-language chain-of-thought. A vision–language policy $\pi_\theta$ (LVLM with parameters $\theta$) defines a distribution over completions, conditioned on the input pair $(x_i, q_i)$ as:

$$o_i = (\hat{a}_i, \hat{y}_i) \in \mathcal{O}, \text{ such that } o_i \sim \pi_\theta(\cdot \mid x_i, q_i) \text{ and } \hat{a}_i = (\hat{c}_i, \hat{T}_i, \hat{w}_i).$$

We denote $\hat{a}_i$ and $\hat{y}_i$ as the associated chain of thought rationale and final answer predicted by a policy, respectively.

**Chart Surrogate Tasks.** Beyond the primary QA objective (i.e., predicting $\hat{y}$), we introduce two *verifiable surrogate tasks* that can be solved as a precursor to reasoning and computing the final output effectively.

- **Chart-Type Prediction.** Identifying the type of chart is a fundamental problem in chart understanding due to significant visual-semantic differences across the types of charts. Predicting the chart type correctly conditions the models to focus on type-specific visual semantics, e.g., for bar graphs - the length of bars, for pie-charts - the sectors of the pie, etc. We therefore predict a discrete type $\hat{c} \in \mathcal{C}$ where $\mathcal{C}$ consists of a fixed set of chart types and compare it to the ground truth $c^*$. Correctly identifying the type guides the model toward type-specific cues and reduces spurious reasoning.
- **Chart-Table Reconstruction.** A chart visualizes data which is structured in the form of the underlying data table $T^* = (C^*, R^*)$ where $C^*$ denotes headers/labels and $R^*$ denotes the row-wise numeric entries. Inferring the data table from the chart is extremely important to ensure accurate reasoning. We reconstruct $\hat{T} = (\hat{C}, \hat{R})$ to condition downstream reasoning on explicit data structure rather than raw pixels. Faithful recovery of $T^*$ is essential, and errors in $\hat{T}$ induce incorrect prerequisites for computing $\hat{y}$. We represent the tables in the JSON format, which consists of two formatted entities - 'columns' containing columns and 'rows' consisting of a list of all rows.

**Proposition 1 (Monotonicity of Conditional Entropy with Chart Surrogates** $(C^*, T^*)$**)** *Let* $(X, Q, C^*, T^*, Y)$ *be random variables for the chart image, query, true chart type, true table, and answer. For any joint distribution over* $(X, Q, C^*, T^*, Y)$,

$$H(Y \mid X, Q) \geq H(Y \mid X, Q, C^*, T^*),$$

*with equality if and only if* $I(Y; C^*, T^* \mid X, Q) = 0$. $H(\cdot)$ *denotes Shannon entropy to quantify uncertainty, while* $I$ *denotes Mutual information* $I(\cdot; \cdot)$ *measures shared information.*

**Proof:** By the nonnegativity of conditional mutual information, $I(Y; C^*, T^* \mid X, Q) = H(Y \mid X, Q) - H(Y \mid X, Q, C^*, T^*) \geq 0$. Rearrange to obtain the inequality; equality holds iff $I(Y; C^*, T^* \mid X, Q) = 0$, i.e., cases where the query can be entirely answered through visual attributes and requires no reasoning. Utilizing accurate chart surrogates helps reasoning. $\square$

## 3.2 GROUP RELATIVE PREFERENCE OPTIMIZATION (GRPO)

GRPO (Shao et al., 2024; Guo et al., 2025) extends the Proximal Policy Optimization (PPO) (Schulman et al., 2017) framework to group-wise preference learning with verifiable rewards. The learning process involves first sampling a *rollout group* on which rewards are computed, followed by a policy update using the GRPO objective as detailed below.

**Rollout groups.** For each $(x_i, q_i)$ we draw a group of $G$ rollouts $\{o_j\}_{j=1}^G \sim \pi_{\text{old}}(\cdot \mid x_i, q_i)$ from a frozen behavior policy $\pi_{\text{old}}$. Each rollout is a completion $o_j = (\hat{a}_j, \hat{y}_j)$ with $\hat{a}_j = (\hat{c}_j, \hat{T}_j, \hat{w}_j)$. Let $\text{tok}(o_j) = (z_1^{(j)}, \ldots, z_{|o_j|}^{(j)})$ be the tokenization of $o_j$, and $z_{<t}^{(j)} = (z_1^{(j)}, \ldots, z_{t-1}^{(j)})$ its prefix.

**Objective.** Within each group, absolute rewards $\{R_j\}_{j=1}^G$ are converted to relative advantages $\hat{A}_j$:

$$\bar{R} = \frac{1}{G}\sum_{j=1}^G R_j, \qquad s_R^2 = \frac{1}{G}\sum_{j=1}^G (R_j - \bar{R})^2, \qquad \hat{A}_j = \frac{R_j - \bar{R}}{\max(1, \; s_R)}. \tag{1}$$

To update the policy $\pi_{old}$ to the new policy $\pi_\theta$, we use a clipped policy surrogate with a sequence-level KL penalty:

$$\mathcal{J}_{\text{GRPO}}(\theta) = \mathbb{E}_{(x_i, q_i) \sim \mathcal{D}} \, \mathbb{E}_{\{o_j\} \sim \pi_{\text{old}}} \left[ \frac{1}{G}\sum_{j=1}^G \frac{1}{|o_j|} \sum_{t=1}^{|o_j|} \left( \min\{\rho_{j,t}(\theta)\,\hat{A}_j, \text{clip}(\rho_{j,t}(\theta), 1-\epsilon, 1+\epsilon)\,\hat{A}_j\} \right) \right.$$

$$\left. - \beta \, D_{\text{KL}}(\pi_\theta \,\|\, \pi_{\text{ref}}) \right], \text{where} \quad \rho_{j,t}(\theta) = \frac{\pi_\theta(z_t^{(j)} \mid x_i, q_i, z_{<t}^{(j)})}{\pi_{\text{old}}(z_t^{(j)} \mid x_i, q_i, z_{<t}^{(j)})}.$$

Here $\epsilon > 0$ is the clipping range, and $D_{\text{KL}}(\pi_\theta \| \pi_{\text{ref}})$ is computed as the average over tokens of $o_j$; $\beta > 0$ weights the KL penalty with respect to the original policy $\pi_{\text{ref}}$.

## 3.3 FORMAT, LENGTH AND ACCURACY REWARD DESIGN

In this section, we discuss the verifiable rewards employed by Chart-RVR. The first set of rewards is commonly utilized GRPO rewards with minor modifications: format, length sufficiency, and answer accuracy. Next, we discuss the rewards around surrogate tasks, namely, chart type and chart-table data construction. Finally, we introduce the Process-Conformity Reward, which ensures the reasoning process does not drift stylistically away from the ground-truth reasoning rationales.

**Format reward.** Regex validation of the two-block schema (`<think>` then `<answer>`) and nested tags of `<type>` enclosing the chart type and `<table>` enclosing the JSON-formatted table representation. We set $R_{\text{fmt}} = 1$ if all format checks pass; 0 otherwise.

**Length (sufficiency) reward.** Let $\ell(o_j)$ be the tokenized length of a rollout $o_j$. Multiple works have (Liu et al., 2025a) observed the impact of reasoning length on CoT traces, wherein longer traces usually improve performance, but overtly long traces begin to overthink and hallucinate, degrading performance. As our reasoning traces are conditioned on the chart type and the underlying data table before actual reasoning, we set the maximum rewards between length thresholds $0 < \eta_1 \leq \eta_2$,

$$R_{\text{len}} = 1, \;\; \text{if } \eta_1 \leq \ell(o_j) \leq \eta_2, \text{otherwise } 0.$$

**Answer accuracy.** The accuracy between the ground truth answer and the predicted answer is calculated on a case-by-case basis, depending on whether the answer consists of textual or numeric outputs. The numeric branch is scale-invariant; textual answers are normalized using norm, which strips trailing special symbols and converts to lowercase. Subsequently, the accuracy is calculated as an exact match for textual outputs, while for numeric outputs, a match within a tolerance $\tau$, usually set to a small number, is calculated, to capture imprecise mathematical values after calculation.

$$
R_{\text{acc}} = \begin{cases} \mathbf{1}\{\text{norm}(\hat{y}) = \text{norm}(y^\star)\}, & \text{textual}, \\ \mathbf{1}\left\{\frac{|\hat{y}-y^\star|}{|y^\star|} \leq \tau\right\}, & \text{numeric}. \end{cases} \tag{2}
$$

Finally, the total rewards for the standard GRPO schema are given as $R_{schema} = R_{fmt}+R_{len}+R_{acc}$

## 3.4 CHART SURROGATE TASK REWARD DESIGN

**Chart Type Prediction.** Given ground-truth chart type $c^*$ and predicted $\hat{c}$, we define the chart type reward as an exact match between the predicted type $\hat{c}$ and ground truth type $c^*$ after normalization:

$$
R_{\text{type}}(\hat{c}, c^*) = \mathbf{1}\big[\text{norm}(\hat{c}) = \text{norm}(c^*)\big],
$$

**Chart Table Reconstruction.** The model emits a JSON table $\hat{T} = (\hat{C}, \hat{R})$, ground truth is $T^* = (C^*, R^*)$. Note that the tables take the form {'columns':{.. , ..},'rows':{[..],[..],...,[..]} }.

$$
R_{\text{table}}(\hat{T}, T^*) = \begin{cases} \underbrace{\frac{1}{|C^*|} \sum_{c \in C^*} \mathbf{1}\{c \in \hat{C}\}}_{\text{column header accuracy}} + \underbrace{\frac{1}{|R^*|} \sum_{r \in R^*} \frac{1}{|r|} \sum_j \mathbf{1}\{r_j = \hat{r}_j\}}_{\text{cell accuracy}}, \\ 0 \qquad \text{otherwise}. \end{cases} \tag{3}
$$

Every correct header contributes $1/|C^*|$; every correct cell adds $1/(|\mathcal{R}^*|\times |r|)$. A parseable JSON yields an additional modest 0.5 reward (to improve reward smoothness) while an unparseable JSON yields $R_{\text{tab}} = 0$, strongly encouraging syntactic validity. The final surrogate task reward can be calculated as: $R_{\text{surr}} = R_{\text{type}} + R_{\text{table}}$.

## 3.5 PROCESS CONFORMITY REWARD DESIGN

Final-answer rewards are sparse and easy to game since models can guess correctly or retrofit a rationale. Recent research has called for evaluating the quality of rationales through a process skeleton (Lee & Hockenmaier, 2025). As a consequence, we propose a process-conformity reward, which incentivizes traces that follow a predefined schema, cite verifiable intermediate quantities, perform the appropriate operations, and remain consistent across steps. This delivers denser credit assignment, discourages hallucinated or decorative CoT, and makes reasoning auditable. It also improves robustness under format/domain shift by enforcing an algorithmic skeleton rather than a dataset-specific style. For chart reasoning, the two primary stages governing the quality of rationale are (i) if it faithfully gathers the *appropriate data* and (ii) reasons with the data sufficiently. We utilize a similarity function, $s$, given a token alphabet $\Sigma$ and a text embedding model $\phi : \Sigma^* \to \mathbb{R}^d$, mapping from natural language text to a $d$-dimensional vector space. Mathematically, for two sentences $a$ and $b$ and cosine similarity $cos$,

$$
s(a, b) = (1 + \cos(\phi(a), \phi(b)))/2; \quad \phi : \Sigma^* \to \mathbb{R}^d; \quad s \in [0, 1]. \tag{4}
$$

**Evidence Gathering Conformity.** For the first stage, we ensure that the data is gathered faithfully. To this effect, we utilize step-wise conformity, where each step is explicitly evaluated to be structurally aligned to the ground truth for the first $m$ steps of each rollout's reasoning (split by steps), denoted as $\hat{w}_{[:m]}$, while ground truth traces are denoted as $w^*_{[:m]}$).

**Reasoning Alignment.** For the second stage, we score the overall reasoning by comparing the model's derivation to the gold steps via text-embedding similarity, encouraging procedural alignment and preventing drift into degenerate traces. The final Process Conformity Reward ($R_{proc}$) is calculated as the sum of $R_{eg}$ and $R_{rs}$. Mathematically,

$$
R_{eg} = \frac{1}{m} \sum_i^m s(\hat{w}_{[:m](i)}, w^*_{[:m](i)}); \quad R_{rs} = s(\hat{w}_{[m:]}, w^*_{[m:]}). \tag{5}
$$

The total Process Conformity Reward $R_{proc}$ is given as $R_{proc} = R_{eg} + R_{rs}$. The final reward $R$ is calculated as a weighted sum of each of the aforementioned Schema Rewards ($R_{schema} \in [0,3]$), Surrogate Task rewards ($R_{surr} \in [0,3]$), and Process Conformity Reward ($R_{proc} \in [0,2]$) where $\lambda_1$ and $\lambda_2 > 0$ are tunable hyperparameters:

$$R = R_{schema} + \lambda_1 R_{surr} + \lambda_2 R_{proc}. \tag{6}$$

## 4 EXPERIMENTS AND RESULTS

### 4.1 DATASET AND MODEL SETTINGS

**Train Datasets.** We utilize ChartQA (Masry et al., 2022), PlotQA (Methani et al., 2020) and ChartFC (Akhtar et al., 2023) datasets to create our CoT datasets. ChartQA consists of a mix of questions based on direct facts and deeper reasoning. PlotQA consists of factoid questions on a completely synthetic dataset. ChartFC consists of yes/no questions requiring deeper reasoning.

**Test Datasets.** We use the ChartQA, PlotQA, and ChartFC test sets as **in-domain** benchmarks. For out-of-domain (OOD) benchmarks, we utilize EvoChart (Huang et al., 2025), which consists of challenging irregular charts in the wild, ChartQAPro (Masry et al., 2025a), which is a more challenging version of ChartQA with more complex charts and questions, and ChartBench (Xu et al., 2023), which consists of questions with complicated reasoning.

**Chart-RVR CoT Reasoning Datasets.** Although the training datasets discussed contain plenty of examples, there is a distinct lack of a reliable source of ground-truth rationales, data tables, and chart-type annotations associated with them. As a consequence, we generate a CoT Chart dataset sampled from the training splits of ChartQA, ChartFC, and PlotQA. To generate faithful CoT rationales, inferring the chart type and generating the associated data tables, we utilize a large-scale SOTA LVLM - Qwen2.5VL-72B. The prompt template for generating the dataset is provided in Figure 3 (Appendix), where both the query and label are provided to the model. (1) *CoT Datasets:* For the CoT dataset, we randomly sample 2,000 datapoints each from the aforementioned datasets based on a specific seed for a total of 6,000 training samples. (2) *CoT-Hard Dataset*: A significant issue in randomly sampling training points from the datasets is the lack of diversity and dominance of easy samples, which constitute queries that require no reasoning, e.g., 'title of the chart. To alleviate this, we specifically sample data from the human-annotated reasoning subset of ChartQA (labeled 'human') and random samples from the ChartFC and PlotQA data. We filter out overly simplistic questions from the dataset for a total of 30,000 training samples.

**Model and Baseline Details:** For all our experiments, we utilize Qwen2.5VL-3B-Instruct (Bai et al., 2025) due to its good benchmark performance and flexibility. Furthermore, to demonstrate the generalizability of Chart-RVR, we use similarly sized Gemma3-3b-it (Team et al., 2025) and InternVL3.5-4B (Wang et al., 2025) LVLMs. We compare our approach to ChartGemma (Masry et al., 2025b), which is the state-of-the-art explainable chart reasoning model, outperforming any other model of similar size (i.e., 3-4 billion parameters). ChartGemma is a fine-tuned model on top of PaliGemma, which outputs an executable Python program as rationales and achieves state-of-the-art performance.

**Training Details. SFT:** We utilize the same system prompt format as in Figure 4(Appendix) for fine-tuning. We train the entire model for 3 epochs, with a learning rate of $1e-5$ for the LLM and projector, while $2e-6$ for the vision tower, with a warm-up ratio of $0.03$. **Chart-RVR:** We utilize TRL's (von Werra et al., 2020) implementation of GRPO with a maximum prompt length of 4096, maximum completion length 768, and number of generations (rollouts) 4 per sample. To further reduce prompt lengths, we utilize the JSON notation to represent the underlying chart tables. For Process Conformity Rewards, we use sentence embeddings using a lightweight embedding model, MiniLM-L6-v2 (Reimers & Gurevych, 2019). We train all models for 4 epochs with a learning rate set as $1e-6$.

### 4.2 EXPERIMENT-0: CHAIN-OF-THOUGHT PROMPTING

First, we evaluate how chain-of-thought prompting affects off-the-shelf LVLMs in Table 1. Although CoT prompting shows gains in LLMs, we observe (Liu et al., 2025b; Xu et al., 2024)

Table 1: Comparison of Direct, CoT, and Structured Prompt.

| Dataset | Direct | CoT | Structured |
|---|---|---|---|
| ChartQA | 82.0 | 41.8 | **73.12** |
| PlotQA | 80.5 | 31.82 | **52.72** |
| ChartFC | 74.4 | 48.02 | **69.20** |
| EvoChart | 48.72 | 18.72 | **29.60** |
| ChartQAPro | 25.7 | 12.01 | **15.80** |
| ChartBench | 66.04 | 29.4 | **51.16** |

an opposite trend in LVLMs, where standard CoT prompting
degrades performance by a large margin compared to direct
prompting. Next, we see how structured prompting (i.e., in-
structing the model to emit chart type, table, and the reasoning
process along with the answer) using the prompt structure shown in Figure 4(Appendix) improves
the results over standard CoT prompting. As can be seen, our structured prompting approach im-
proves performance, but is still significantly less than direct prompting.

Table 2: Main benchmark results across 6 diverse chart datasets. The 'Exp?' column signifies if
the approach is explainable (i.e., outputs CoT rationales or equivalent, like Python programs). We
observe that Chart-RVR-3B and Chart-RVR-3B-Hard achieve benchmark performance across all
benchmarks as compared to SFT and chart-specific models. The performance improvement is more
pronounced on out-of-domain (OOD) datasets, as signified in the last 3 columns.

| Approach | Exp? | ChartQA | PlotQA | ChartFC | EvoChart | ChartQAPro | ChartBench |
|---|---|---|---|---|---|---|---|
| **Direct Prompting** | | | | | | | |
| Q2.5VL-Ins | ✗ | 82.0 | 80.5 | 74.4 | 48.72 | 25.7 | 66.04 |
| **Explainable Models with Rationales** | | | | | | | |
| Q2.5VL-Ins (CoT) | ✔ | 73.12 | 52.72 | 69.20 | 29.6 | 15.80 | 51.16 |
| ChartGemma | ✔ | 76.44 | 33.28 | 70.33 | 36.96 | 10.93 | 40.56 |
| **Fine-tuned Models with Rationales** | | | | | | | |
| Q2.5VL-SFT | ✔ | 83.08 | 74.18 | 77.30 | 46.08 | 23.56 | 64.64 |
| Q2.5VL-Ins (A+F+L) | ✔ | 76.72 | 56.22 | 58.58 | 38.88 | 17.55 | 48.1 |
| Q2.5VL-Ins (A+F+L+Tasks) | ✔ | 81.8 | 76.24 | 63.85 | 51.68 | 27.66 | 65.28 |
| **Chart-RVR-3B** (Ours) | ✔ | 84.56 | **78.68** | 77.62 | 53.36 | 28.38 | 68.32 |
| **Curated Data Fine-tuned Models with Rationales** | | | | | | | |
| Q2.5VL-SFT-Hard | ✔ | 84.28 | 75.54 | 77.90 | 49.36 | 23.20 | 65.12 |
| **Chart-RVR-3B-Hard** (Ours) | ✔ | **85.76** | 77.9 | **80.07** | **54.24** | **28.64** | **69.46** |

## 4.3 EXPERIMENT-1: BENCHMARK PERFORMANCE

We report the benchmark performance of Chart-RVR-3B and Chart-RVR-3B-Hard (trained on the
CoT-Hard dataset) in Table 2 on six diverse datasets. In addition, we also compare our approach with
Direct Prompting approaches on the same base model, even though they are non-explainable. We
observe that Chart-RVR consistently outperforms all approaches, including the chart-specific base-
line ChartGemma. The improvements over SFT on in-domain datasets are approximately 1-2%.
However, significant improvements are observed on Out-of-Domain datasets EvoChart (**+7.28%**),
ChartQAPro (**+4.82%**), and ChartBench (**+3.68%**). In addition, Chart-RVR-3B-Hard boosts per-
formance by an extra 1–2% across the board, highlighting the effectiveness of high-quality data
curation.

**Impact of Various Rewards.** Next, we discuss ablation studies with respect to the proposed re-
wards. As standard GRPO implementations usually only optimize format, length, and accuracy
rewards, we report the results in Table 2 depicted as Q2.5VL-Ins (A+F+L). Next, we report perfor-
mance with all formats, accuracy, length, and surrogate task rewards Q2.5VL-Ins (A+F+L+Tasks).
As can be observed, standard GRPO's learning process is worse than SFT, implying **naive GRPO is
ineffective**. Once surrogate tasks are introduced (A+F+L+Tasks), the reasoning process beats SFT
on multiple datasets. Finally, adding the Process Conformity Reward makes our method exceptional
on all benchmarks, highlighting its utility.

## 4.4 EXPERIMENT-2: CHART-RVR VERSUS SFT PERFORMANCE COMPARISON

To demonstrate the consistent gains of Chart-RVR over Supervised Fine Tuning (SFT), we ran-
domly sample the CoT-dataset using 3 different seeds, with 6,000 samples from the training set of
ChartQA, PlotQA, and ChartFC. We train three separate SFT and Chart-RVR models, each on the
various training splits using the same prompt structure as Figure 4(Appendix). We report the aver-
age performance and standard deviation in Table 3a over all 3 seeds. As can be seen, Chart-RVR

Table 3: (a) Chart-RVR's consistent improvements over SFT. (b) Surrogate Task Performance.

(a) Comparison between fine-tuned models using 3 distinct seeds of the ChartRVR-CoT Dataset. Chart-RVR improves performance over SFT across all datasets, with OOD improvements more pronounced.

(b) Performance across different datasets on the surrogate tasks: (i) Chart Type Accuracy (Type Acc) and (ii) Underlying table creation (Tab). Note that the Tab is the Edit-Distance errors, where lower is better.

| Dataset | Direct | CoT | SFT | Chart-RVR |
|---|---|---|---|---|
| ChartQA | 82.0 | 73.12 | 83.18 ± 0.33 | **83.87 ± 0.68** |
| PlotQA | **80.50** | 52.72 | 76.05 ± 1.65 | **78.71±0.20** |
| ChartFC | 74.4 | 69.20 | 76.67 ±0.54 | **78.08±1.36** |
| EvoChart | 48.72 | 29.6 | 46.50 ± 0.36 | **52.16±0.86** |
| ChartQAPro | 25.7 | 15.80 | 24.52 ±0.48 | **28.30±1.00** |

| Dataset | CoT | | SFT | | Chart-RVR | |
|---|---|---|---|---|---|---|
| | Acc (↑) | Tab (↓) | Acc (↑) | Tab (↓) | Acc (↑) | Tab (↓) |
| ChartQA | 0.87 | 0.46 | 0.94 | **0.38** | **0.95** | 0.49 |
| PlotQA | 0.70 | 0.65 | **0.78** | 1.13 | 0.77 | **1.03** |
| ChartFC | 1.00 | 0.65 | 1.00 | 0.20 | 1.00 | 0.20 |
| EvoChart | 0.74 | 0.99 | 0.81 | 1.28 | **0.84** | **1.22** |
| ChartQAPro | 0.69 | 0.72 | 0.72 | 1.05 | 0.72 | 1.05 |

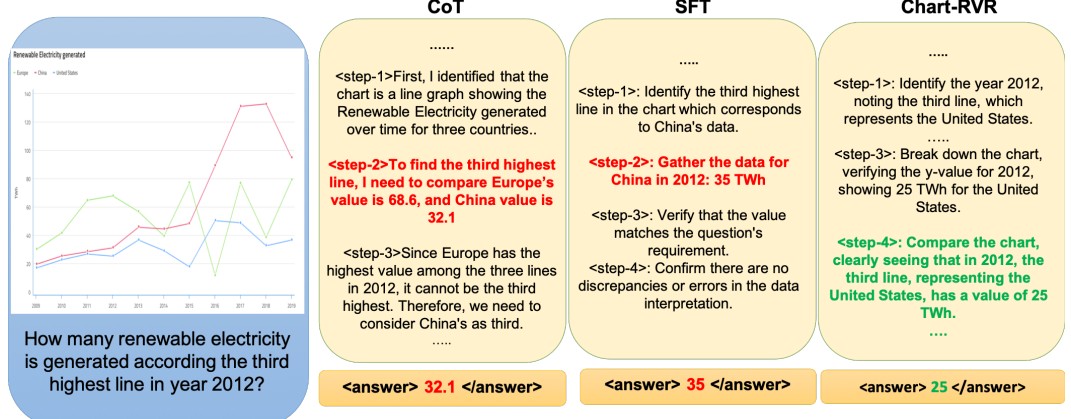

(a) CoT gets the initial data gathering step wrong, attributing the green line 14%, which is incorrect, compromising the entire reasoning process, while SFT fails similarly on Step 2, wherein it wrongly attributes the value of 14 instead of 21. Chart-RVR reasons faithfully by smaller, more accurate steps to output the correct answer.

(b) Both CoT and SFT fail in the initial steps by misidentifying the third-highest category (Europe and China, respectively) and the relevant line, while Chart-RVR correctly recognizes the US as the third-highest.

Figure 1: **Chain-of-thought rationales on EvoCharts (OOD).** We demonstrate CoT rationales for Structured prompting on base model, SFT model, and Chart-RVR. We highlight the mistake in a particular reasoning step in red font. See Appendix for additional examples.

consistently outperforms SFT across both ID and OOD datasets by **1–2%** and **4–6%** respectively, demonstrating the efficacy of our method over SFT. In addition, the average results are at par with the results on a single CoT dataset split (as shown in Table 2), highlighting robust convergence.

## 4.5 EXPERIMENT-3: RESULTS ON SURROGATE TASKS

In Table 3b, we report the performance on the surrogate tasks as discussed in Section 3.4. Note that the chart-type accuracy measures how accurately the type of chart is predicted, while for table reconstruction, we measure Edit Distance errors as defined between predicted table $\{\hat{R}, \hat{C}\} \in \hat{T}$ and

ground truth table $\{R^*, C^*\} \in T^*$ as:

$$E_T(\hat{T}, T^*) = (1/|C^*|) \sum_{c \in C^*} \mathbf{1}[c \notin \hat{C}] + (1/|R^*| \times 1/|r|) \sum_{r \in R^*} \sum_j \mathbf{1}[r_j \neq \hat{r}_j], \qquad (7)$$

We observe that SFT and Chart-RVR moderately boost chart-type accuracy, implying that the base model is already decent at the chart-type identification task. However, our method improves underlying data table reconstruction by 0.06 points on EvoChart (OOD) as compared to using SFT.

## 4.6 EXPERIMENT-4: INTERPRETABILITY ANALYSIS AND QUALITY OF RATIONALES

To measure the quality of chain-of-thought rationales output by our method, we design the *Explainable Information Gain* metric ($\Delta \log P$), which measures the difference in the probability of predicting the ground truth answer $y^*$ given the image $x$ and the output CoT rationale $a$ using an oracle model $W$. We utilize the Qwen2.5VL-72B, a SOTA LVLM, as the oracle. Intuitively, our metric measures the additional *information* added by the rationales contributing to the **certainty** of the answer. Mathematically,

$$\Delta \log P = \log P_{\mathrm{W}}(y^\star | x, a) - \log P_{\mathrm{W}}(y^\star | x) \qquad (8)$$

We report $\Delta \log P$ for correctly predicted responses in Table 4 for CoT, SFT, and Chart-RVR. We omit ChartFC (ID) and ChartBench (OOD) due to overwhelming binary questions. Surprisingly, not only is CoT unable to provide accurate answers, but the CoT rationales are also unhelpful for improving explainability by reducing the certainty of the correct answer. Next, SFT and Chart-RVR both improve explainability, but our method outperforms par-

Table 4: Explainability Results.

| Dataset | $\Delta$ CoT | $\Delta$ SFT | $\Delta$ Chart-RVR |
|---|---|---|---|
| ChartQA | -5.04 | **-2.22** | -4.3 |
| ChartQA (human) | -3.46 | +0.09 | **+0.3** |
| PlotQA | -2.25 | +1.13 | **+3.66** |
| EvoChart | -9.13 | -6.82 | **+0.02** |
| ChartQAPro | -0.04 | +1.75 | **+2.41** |

ticularly on OOD datasets. Interestingly, all methods degrade as compared to direct prompting on ChartQA, implying some memorization in the base model. However, on the deeper/harder reasoning samples, ChartQA (human), both SFT and Chart-RVR improve. These results are a testament to the improved explainability of Chart-RVR. A visual example is demonstrated in Figure 1, comparing CoT, SFT, and Chart-RVR. Additionally, we also utilize an additional Oracle model to judge the rationales similarly via LLaVA-Next-72B in Table 8 (Appendix). Finally, we also conduct a human study wherein Chart-RVR reasoning is more interpretable than those of SFT and CoT, details and results of which are shown in Figure 9 (Appendix).

## 4.7 EXPERIMENT-5: CHART-RVR APPLIED TO DIVERSE LVLM ARCHITECTURES

In Table 5, we report the performance on the 3 OOD benchmarks using two different backbone VLMs: Gemma3 and InternVL3.5. We observe that **SFT** provides a reasonable baseline improvement over CoT, with InternVL generally outperforming Gemma across all datasets. To demonstrate the efficacy of Chart-RVR, we compare against **GRPO (A+F+L+Tasks)**, which improves over SFT on Gemma models, but shows inconsistent gains on InternVL. Finally, **Chart-RVR** achieves the best overall performance, yielding consistent improvements across both Gemma and InternVL on all benchmarks except ChartQAPro where it is at par. These results indicate stronger generalization capabilities compared to both SFT and standard GRPO with Surrogate tasks.

Table 5: **Ablation across different training settings on OOD benchmarks.** Columns are grouped by dataset with subcolumns for Gemma-3 and InternVL-3.5 (IVL) respectively. Chart-RVR outperforms across all datasets and models.

| Setting | EVOCHART | | CHARTQAPRO | | CHARTBENCH | |
|---|---|---|---|---|---|---|
| | Gemma | IVL | Gemma | IVL | Gemma | IVL |
| CoT | 21.04 | 45.36 | 24.82 | 15.91 | 42.28 | 54.42 |
| SFT | 21.92 | 44.08 | 23.56 | **29.34** | 54.14 | 63.16 |
| GRPO (A+F+L+Tasks) | 42.26 | 50.04 | 29.51 | 20.43 | 53.18 | 64.62 |
| **Chart-RVR (Ours)** | **42.48** | **50.24** | **32.59** | **29.34** | **58.18** | **64.78** |

## 5 CONCLUSION

In this paper, we introduced Chart-RVR, a general-purpose reinforcement learning framework for explainable chart reasoning built on verifiable rewards. Our method augments standard GRPO with rewards for chart surrogate tasks, i.e., chart-type prediction and table reconstruction, as well as a process-conformity objective that encourages faithful, step-by-step reasoning aligned with ground-truth rationales. Empirically, Chart-RVR improves both answer accuracy and explanation quality across six benchmarks and three LVLMs (Qwen2.5VL, Gemma3, and InternVL-3.5), with the largest gains under distribution shift (EvoChart, ChartQAPro, ChartBench), indicating stronger OOD generalization than SFT and vanilla GRPO. We also demonstrate the improved explainability of the CoT traces produced by Chart-RVR.

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

# A  APPENDIX

## A.1  SALIENT DATASET CONSTRUCTION DETAILS

**Chart Types ($\mathcal{C}$).** Even though a large variety of chart types exist, most of the commonly found charts can be categorized into 10 fundamental categories. We instantiate a controlled set of chart families: *Line*, *Bar*, *Stacked Bar*, *Pie/Donut*, *Histogram*, *Scatter*, *Area*, *Stacked Area*, *Bubble*, and *Treemap*. [1]

**Construction.** To generate the Chart-RVR CoT datasets, we utilize a large open-source LVLM as an oracle namely Qwen2.5VL-72B-Instruct (Bai et al., 2025). For the Chart-RVR datasets, the prompt template utilized is shown in Figure 3.

1. *Normalization.* Images are resized with aspect-ratio preservation to a minimum of 512 leading edge size; tables are canonicalized to a header (columns) + rectangular body (rows) with string conversion where applicable. We map the dataset's native chart label to the chart types above.
2. *Templated prompting.* We instantiate Figure 3 with (a) the canonical chart label, (b) a compact task description, and (c) formatting constraints (typed tags, JSON schema, operation tokens). The template explicitly separates (i) *data gathering* steps from (ii) *computation* steps to enable process supervision.
3. *Sampling and pre-screen.* For each (image, question) pair, we output rationales via deterministic inference with temperature set to 0. Candidates failing structural checks (JSON parse, length bounds, required tags) are discarded before reward computation.

**Manual Filtering of sub-par rationales.** Although our rewards prune many low-quality generations automatically, we perform a light manual pass to remove pathological rationales that would otherwise pollute training data points. A rationale is marked *sub-par* and discarded if any of the following holds:

---

[1] We normalize synonyms to a canonical label: *Column→Bar*, *Donut→Pie*, *Point→Scatter*. Grouped bars are labeled *Bar*; cumulative variants are *Stacked Bar*.

1. *Unparseable structure:* the emitted JSON blocks (table tag) fail to parse or violate schema (missing keys, non-rectangular rows).
2. *Data hallucination:* The rationale cites a category/series not present in the ground-truth table, i.e., non-sensical characters, special symbols, etc.
3. *Length:* We filter out all rationales with fewer than 3 steps of reasoning.

We utilize the following procedure for final dataset validation:

1. For pass 1, the prompt structure shown in Figure 3 is used to generate the chain-of-thought data.

2. For pass 2, all samples with overtly short/long and incorrect reasoning traces are filtered (i.e. trace length less than 3 or more than 8 based on manual inspection and outlier rejection). To validate if a trace is wrong, we algorithmically check if the last CoT line trace contains the correct answer. A total of 358 samples are filtered out in this step out of 15000.

3. For pass 3, three doctoral-level researchers are employed to validate the reasoning traces of a randomly sampled data subset (1000 samples). Each researcher filtered fewer than 6 samples (1%) in total, with a 100% agreement between them.

We will release the filtered dataset upon acceptance for transparency and faithful reproduction. Additionally, the smaller CoT Dataset with appropriate seeds will also be released.

**Agreement of annotation done using human validation** We manually validated 100 randomly sampled oracle-annotated charts from the EvoChart and ChartQAPro datasets (note no ground truth table annotations are available for either). We categorize each data sample into 2 categories - (1) Needs Approximation? - which entails guessing the correct values from the chart axes rather than labelled counterparts, "Factoid/Direct" - which entails looking at axes marks to determine the correct values. In Table 6, we report the Kendall's Tau: the inter-annotator agreement score b/w human (Annotator-1) and Oracle model (Annotator-2). We also show examples of the categorization in Figure 2.

| Category | #Samples | Cohen's $\kappa$ |
|---|---|---|
| Needs Approximation | 48 | 0.91 |
| Factoid/Direct | 152 | 0.98 |
| **Overall** | 200 | 0.97 |

Table 6: Inter-annotator agreement between human annotators and the Oracle model.

**Emphasizing OOD benchmarks.** Many widely used chart reasoning datasets like ChartQA, PlotQA, and ChartFC have been repeatedly incorporated (in whole or in part) into instruction-tuning corpora, synthetic data expansions, and public multimodal mixtures that contemporary LVLMs are trained or aligned on, together with related datasets such as FigQA and DVQA[2]. This creates a realistic risk of distributional familiarity and data leakage at pretraining/finetuning time, yielding optimistic "in-distribution" estimates on these suites. To assess generalization beyond such exposure, we therefore evaluate on 'truly' out-of-distribution (OOD) results on EvoChart, ChartBench, and ChartQA-Pro, which were not used in model pretraining or alignment and contain chart styles, templates, and question programs that differ from the legacy sets. As shown in later sections, it can be observed that datasets like EvoChart possess significantly harder chart images than benchmark sets like ChartQA. Throughout the paper, we treat ChartQA/PlotQA/ChartFC as 'ID' baselines and use EvoChart/ChartBench/ChartQA-Pro to measure robustness, reasoning fidelity, and explainability under genuine distribution shift. Multiple present chart works do not have a truly OOD benchmarking evaluation suite. Truly generalizable chart reasoning with explanations remains an open problem for frontier LVLMs.

---

[2]Note: Because most LVLM training mixtures are only partially disclosed, we cannot exhaustively audit every provider. Our designation reflects (i) the public availability and long shelf-life of ChartQA/PlotQA/ChartFC, (ii) their documented use in multiple open instruction-tuning recipes as documented in InternVL's technical report (Chen et al., 2024)

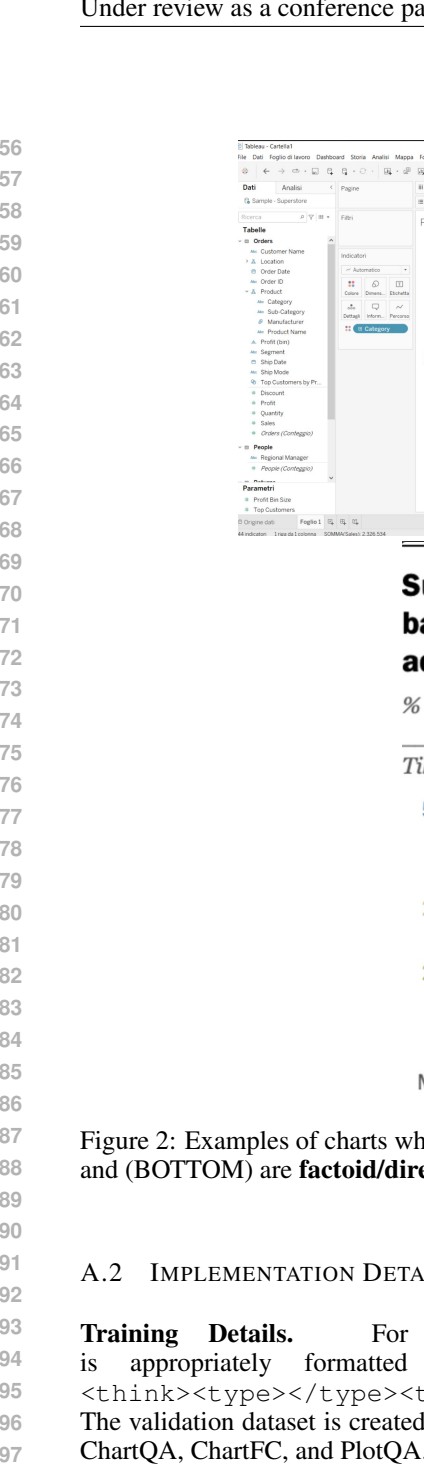

Figure 2: Examples of charts where the Oracle and human annotations (TOP) **need approximation** and (BOTTOM) are **factoid/direct**; the corresponding annotated data tables may differ.

## A.2 IMPLEMENTATION DETAILS

**Training Details.** For SFT, each training sample's CoT trace and answer is appropriately formatted into the expected prompt format and adhering to `<think><type></type><table></table>...</think><answer></answer>`. The validation dataset is created using **500** randomly sampled data points from the training sets of ChartQA, ChartFC, and PlotQA.

- **SFT:** For SFT, we train the model for a maximum of 3 epochs and select the model with the highest accuracy on the validation set. Additionally, we utilize the AdamW optimizer for training with a training batch size of 4 across 2 NVIDIA H100 GPUs. The learning rate schedule utilized is linear with a maximum learning rate of $1e-5$ and 1000 warmup steps. During training, the entire model is trained with FP16 precision, and the vision tower is trained using an LR of $2e-6$.

- **Chart-RVR:** For Qwen2.5VL models, we utilize a maximum completion length of 768 and a total prompt length of 4096. For Gemma and InternVL models, due to more efficient vision token computations, we utilize a maximum prompt length of 3072 with a maximum completion length of 512. For Qwen2.5VL, we utilize a maximum learning rate of $1e-6$ while for Gemma and InternVL, the learning rate is reduced to $5e-7$ and $2e-7$ respectively. The training process takes place on 4 NVIDIA H100 GPUs with a per-device batch size of

---

**System Prompt template for CoT Dataset Generation**

You are helping me answer questions on charts.
You have to look both at the chart picture and the question.
The question and the answer will be provided to you.
First you have to recover the table data from the chart image in JSON format.
For the chart image, output only a JSON object with:
"columns": list of column headers,
"rows": list-of-lists, one per data row
No prose, no comments.
1. Respond with **only** a JSON object inside a "'json code fence.
2. The JSON must use exactly this schema:
{
"columns": [...],
"rows": [...]
}
3. Do NOT output HTML, Markdown, or commentary. Any deviation gets zero reward.
Next, think step by step in as many small steps as required to answer the question based on the chart.
Lastly, also predict the type of chart out of the following:
"line", "bar", "stacked bar", "pie", "histogram", "scatterplot", "area", "stacked area", "bubble", "treemap"
Format:
### Question: <question>
### Answer: <answer>
### Table: <json table>
### Reasoning:
<step-1>: Provide a description of reasoning
<step-2>: Gather ALL the appropriate data from the chart
<step-3>: Break down the query into smaller parts and verify each part with the data
...
<step-n>: Do the final calculation or reasoning to derive the answer
<step-n+1>: VERIFY the final answer is correct for no hallucinations
### Type: <type of chart>

---

Figure 3: **System prompt template** used across Structured CoT, SFT, GRPO, and Chart-RVR.

2. The number of rollouts is set to 4. The total number of epochs is set at 4 for all models and configurations.

**Hyperparameter Tuning.** Note that the Chart-RVR setup has 3 major hyperparameters to tune, which assign relative weights to each component of the reward design. We set $\lambda_1$, i.e., the surrogate task reward weight, to be $0.5$, while $\lambda_2$, i.e., the process conformity reward weight, is set to 1. Finally, the hyperparameter $\alpha$, which balances the contribution of the two components of the process conformity reward, is chosen to be 2 for Qwen, Gemma, and InternVL models to assign more weight to the actual reasoning. We observe that due to smooth normalization, minor changes to the values of $\lambda_1, \lambda_2$, and $m$ do not affect performance significantly. The value of $m$ is set to 3. The parameters $\eta_1, \eta_2$ are set at 150 and 400, respectively. We report some training curves demonstrating the behavior of different reward values in Figure 5.

**Evaluation Setup.** For all our evaluations, we utilize the public test sets of the datasets. For ChartQA[3], ChartQAPro[4], EvoChart[5], and ChartFC[6], we utilize the entire test sets sampled from

---

[3]https://huggingface.co/datasets/HuggingFaceM4/ChartQA
[4]https://huggingface.co/datasets/ahmed-masry/ChartQAPro
[5]https://github.com/MuyeHuang/EvoChart
[6]https://github.com/mubasharaak/ChartCheck

sources as listed in the footnotes. For ChartBench[7] and PlotQA[8], due to their massive sizes, we sample a 5000-sized subset using a fixed seed. For all models, we resize the leading image edges to a minimum and maximum of 448 and 812, respectively, using Bicubic interpolation. For all structured CoT, SFT, and GRPO evaluations, we utilize the prompt template in Figure 4 as the system prompt for all models.

**Evaluation Metrics.** For all datasets, we utilize the Relaxed Accuracy metric as proposed in (Masry et al., 2022) and commonly utilized in subsequent chart works (Liu et al., 2023; Masry et al., 2025b), etc. Relaxed Accuracy considers the predicted numerical answers within a tight threshold as correct (not an exact match). The threshold value is set as 0.05 (5%). Mathematically for a prediction $\hat{y}$ and ground truth $\hat{y}*$,

$$\text{Relaxed Accuracy}(\hat{y}, y*) = \mathbf{1}\left[\frac{|y* - \hat{y}|}{y*} \leq 0.05\right]$$

, where $\mathbf{1}$ is the Indicator Function. Note that an exact match is utilized for non-numeric or mixed alphanumeric answers. For ChartFC, as all the questions have a binary Yes/No answer, we append the line 'Answer Yes/No' in the prompt template to elicit faithful responses and suppress True and False outputs, which, in theory, answer the question correctly.

---

**System Prompt template for Structured CoT / SFT / GRPO / Chart-RVR**

You are a vision-language assistant. You are given a chart image and a query about the chart. Think step-by-step about how to answer the query based on the chart image and then provide the final answer.
### Output format:
Respond **with exactly two blocks in order and nothing else**:
<think>
First output the type of chart in <type>, then output the underlying data table and finally, think step-by-step about how to answer the query based on the chart image and then provide the final answer. <type> Type of chart - one word from line, bar, stacked bar, pie, histogram, scatterplot, area, stacked area, bubble, treemap. </type>
Next output the data table in the <table></table> tags
<table>
json table - for the chart image, output only a JSON object with: "columns": list of column headers, "rows": list-of-lists, one per data row
No prose, no comments.
1. Respond with **only** a JSON object
2. The JSON must use exactly this schema: { "columns": [...], "rows": [[...], [...],..., [...]] }
3. Do NOT output HTML, Markdown, or commentary. Any deviation gets zero reward.
</table>
Provide your reasoning here in steps:
<step-1>: Provide a description of reasoning
<step-2>: Gather ALL the appropriate data from the chart
<step-3>: Break down the query into smaller parts and verify each part with the data
...
<step-n>: Do the final calculation or reasoning to derive the answer
</think>
<answer>
Final answer on a single line
</answer>

Figure 4: **System prompt template** used across Structured CoT, SFT, GRPO, and Chart-RVR.

---

[7]https://huggingface.co/datasets/SincereX/ChartBench
[8]https://github.com/NiteshMethani/PlotQA/blob/master/PlotQA_Dataset.md

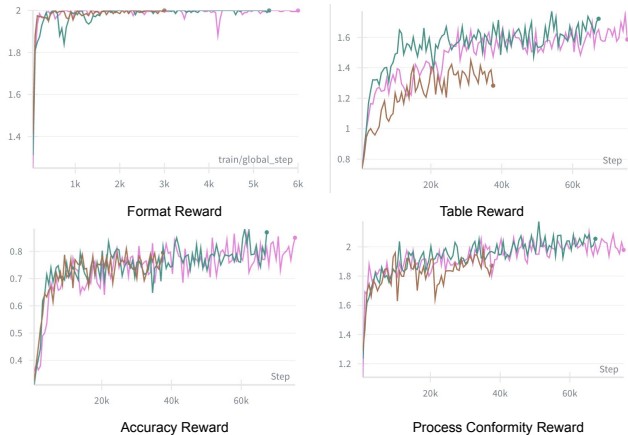

Figure 5: Chart-RVR Reward maximization during training on 3 separate CoT datasets on Qwen2.5VL-3B.

### A.3 PRACTICAL REWARD HACKING AND MITIGATION STRATEGIES

Reward hacking is a common phenomenon in Reinforcement Fine-Tuning (RFT), In this section, we demonstrate common reward-hacking behavior observed in our experiments and mitigation strategies employed by us to alleviate these concerns. Note that the reward hacking behavior is usually observed during the early stages of the training and can cause a sudden training collapse, from which the policy never recovers.

**Stacked Rewards on Length.** As the models are incentivized to output longer traces (rollouts) via the length reward, in some cases, this results in unrelated characters being outputted to 'fill' in the extra tokens by new line tokens ('\n') or repeating multiple identical redundant steps. This is often observed when smaller LVLMs, which do not output their CoT traces reliably, are suddenly incentivized to output much longer traces. If left unmitigated, this can cause a training collapse where the model can never recover from this particular policy, which technically still maximizes the reward. To alleviate this, we utilize a stacked reward design, where a partial reward is assigned for reasoning lengths above certain thresholds. This treats the length reward maximization as a 'warm' start process and nudges the model to output longer traces gradually and not immediately upon the start of training. We utilize a 0.5 reward for token lengths of 100 or more, and finally, the full 1.0 is set when the length exceeds $\eta_1$ tokens. Additionally, we also penalize 'filler' tokens by checking if more than 5 '\n' occur in contiguous tokens; we provide a 0 reward.

**Stacked Rewards on Table Reconstruction.** As detailed in the main text, we utilize the Table reconstruction rewards in 3 tranches - a successful JSON parse of the tokens inside each rollout's <table> tags gives a 0.5 reward. As the reward function on table reconstruction is extremely dense, we warm start the process by assigning a 0.5 extra reward if 'columns' and 'rows' appear in the rollout's JSON parsing.

### A.4 TRAINING CURVES AND REWARD MAXIMIZATION BEHAVIOR

In Figure 5, we visually demonstrate the reward maximization during the Chart-RVR training process on 3 splits of the CoT dataset. We observe that the format rewards are maximized very early in the training with minimal changes observed throughout the process. The accuracy reward gradually improves and stagnates after about 60k steps (about 2 epochs). Similarly, the Table Rewards and Process Conformity Rewards demonstrate a smooth increase initially and then are maximized around the same number of steps. All rewards show a massive improvement during early training.

## A.5 VISUAL COMPARISON BETWEEN COT, SFT, AND CHART-RVR

In this section, we demonstrate visual examples where Chart-RVR outperforms both CoT and SFT. First, we demonstrate Chart-RVR's effectiveness on OOD data on the EvoChart benchmark in Figure 6. As can be seen, CoT and SFT misidentify the number of categories as they only look at the legend on the left, while Chart-RVR correctly counts the number of colors. In the second case, Chart-RVR correctly identifies the value of the second April, which is a duplicate entry on the x-axis.

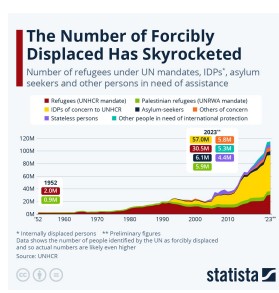

**Query:** ['How many refugee categories were mentioned in the chart?']
**Ground Truth:** ['7']

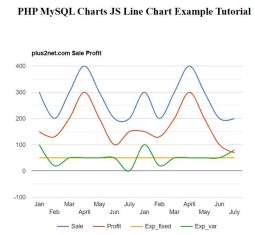

**Query:** ['What is the value of the red line at the second April?']
**Ground Truth:** ['300']

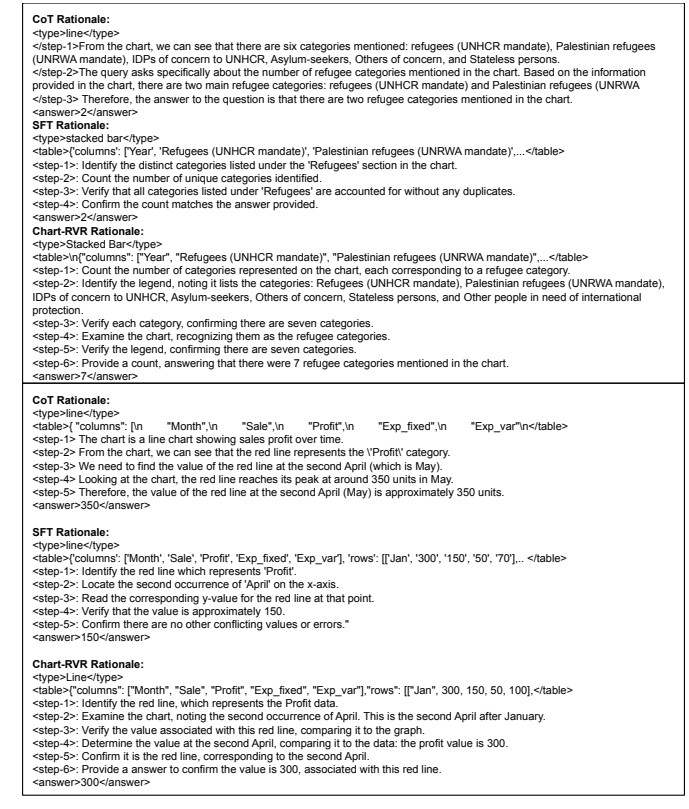

Figure 6: Example failure cases from the EvoChart dataset (OOD) where Chart-RVR outperforms CoT and SFT on 2 challenging examples.

Next, in Figure 7, Chart-RVR correctly identifies the 'BeReal' category, which is not picked up by both CoT and SFT, as it is only one data point on the line chart. This reinforces the need to condition reasoning on faithful table reconstruction. In the next image, Chart-RVR is able to capture the entity '3.6' accurately, where both CoT and SFT fail.

In Figures 10,11, and 12, we report cases where Chart-RVR outperforms both CoT and SFT on the ChartQA dataset. We observe that Chart-RVR is particularly accurate in cases where the chart is extremely complex.

## A.6 ADDITIONAL RESULTS ON CHARXIV

We report more results on the CharXiv-Reasoning benchmark Wang et al. (2024) in Table 7.

| Method | CharXiv (reasoning) |
| --- | --- |
| CoT | 26.2 |
| SFT-Hard | 29.6 |
| Chart-RVR-Hard | **34.3** |

Table 7: Performance on the CharXiv reasoning subset.

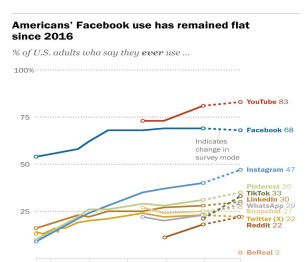

**Query:** ['How many apps are mentioned in the chart?']
**Ground Truth:** ['11']

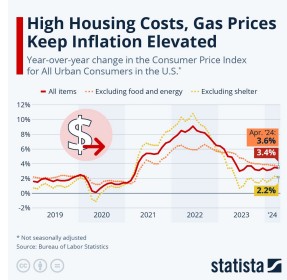

**Query:** ['What is the percentage value of excluding food and energy in April 2024?']
**Ground Truth:** ['3.6']

**CoT Rationale:**
<type>line</type>
<table>{ "columns": ["App","Year"],"rows": [ ["YouTube", "2016"], ["Facebook", "2016"], ["Instagram", "2016"],["Pinterest"
<step-1>The chart is a line graph showing the percentage of U.S. adults who say they ever use various social… will count the number of distinct apps mentioned in the legend at the top of the chart.\n</step-2>\n<step-3>\nI will verify that all the apps listed in the legend are indeed mentioned in the data rows.\n</step-3>\n<step-4>\nThe apps mentioned in the chart are YouTube, Facebook, Instagram, Pinterest, LinkedIn, Snapchat, Twitter (X), Reddit, and BeReal
<answer>9</answer>
**SFT Rationale:**
<type>line</type>
<table>\n{'columns': ['Year', 'YouTube', 'Facebook', 'Instagram', 'Pinterest', 'LinkedIn', '...</table>
<step-1>: Provide a description of reasoning  The chart lists various social media platforms along with their usage percentages over
<step-2>: Gather ALL the appropriate data from the chart  There are ten distinct lines representing different social media platforms: YouTube, Facebook, Instagram, Pinterest, LinkedIn, WhatsApp, Snapchat, Twitter (X), Reddit, and BeReal.\n\n
<step-3>: Break down the query into smaller parts and verify each part with the data  The question asks for the number of apps mentioned. By counting the unique lines in the chart, we can determine there are 10 distinct platforms.\n\n
<step-4>: Do the final calculation or reasoning to derive the answer  Counting the lines confirms that there are indeed 10 apps
<answer>10</answer>
**Chart-RVR Rationale:**
<type>Line</type>
<table>{"columns": ["Year", "YouTube", "Facebook", "TikTok", "Instagram", "Pinterest", "LinkedIn"...</table>
<step-1>: Count the number of app names on the chart, identifying them along the legend.
<step-2>: Gather information about each app, recognizing it on the graph, which represents YouTube, Facebook, TikTok, Instagram, Pinterest, LinkedIn, WhatsApp, Snapchat, Twitter (X), Reddit, and BeReal.
<step-3>: Verify the chart, counting each app mentioned.
<step-4>: Examine the data, confirming there are 11 apps in total.\n<step-5>: Verify the legend, confirming it lists all the applications.\n<step-6>: Provide a count, answering that there are 11 apps mentioned in the chart.
<answer>7</answer>

**CoT Rationale:**
<type>scatterplot</type>
<table>\n{\n  "columns": [\n    "Year",\n    "All items",\n    "Excluding food and energy",\n    "Excluding shelte…</table>
<step-1> The chart shows the year-over-year change in the Consumer Price Index (CPI) for all urban consumers in the U.S. over time.</step-1>
<step-2> From the data provided, we can see that the CPI excluding food and energy was at 2.2% in April 2024.</step-2>
<step-3> This information directly answers the question asked.
<answer>2.2</answer>

**SFT Rationale:**
<type>line</type>
<table>{'columns': ['Year', 'All items', 'Excluding food and energy', 'Excluding shelter'], 'rows': ['2019', 'None',</table>
<step-1>: Identify the relevant data point for April 2024 under the 'Excluding food and energy' category.
<step-2>: The data shows that the percentage value for 'Excluding food and energy' in April 2024 is 3.4%.
<step-3>: Verify that there are no other conflicting values or additional context needed for this specific question.
<step-4>: Confirm that the answer matches the provided answer of 3.4%.
<step-5>: VERIFY the final answer is correct for no hallucinations.
<answer>3.4</answer>

**Chart-RVR Rationale:**
<type>Line</type>
<table>{"columns": ["Year", "All items", "Excluding food and energy", "Excluding shelter"],"rows": [["2019", 2, 2, 2], …</table>
<step-1>: Examine the chart, identifying the year for April 2024, which is represented as \'24\'.
<step-2>: Gather the value associated with \'Excluding food and energy\' for this year. It represents the data for April 2024.
<step-3>: Verify the percentage value, confirming it is 3.6%.
<step-4>: Compare the chart with the information given, noting the value associated with April 2024.
<step-5>: Confirm that this corresponds to the \'Excluding food and energy\' category, with a value of 3.6%.
<step-6>: Provide a direct answer to confirm the question asks for this percentage value in April 2024.
<answer>3.6</answer>

Figure 7: Example failure cases from the EvoChart dataset (OOD) where Chart-RVR outperforms CoT and SFT on 2 challenging examples.

## A.7 ADDITIONAL EXPLAINABILITY RESULTS

Table 8 reports the improvement on the explainability metric for LLaVA-Next-72B under three training modes (CoT, SFT, Chart-RVR). Chart-RVR delivers the largest gains on 4/5 datasets, notably +0.38 on ChartQA and +0.23 on EvoChart—while SFT narrowly leads on PlotQA (+0.1642 vs +0.1632). Overall gains are modest on harder sets (e.g., ChartQAPro = +0.058), but Chart-RVR provides the most consistent uplift in explainability.

Table 8: Explainability Results using LLaVA-Next-72B.

| Dataset | $\Delta$ CoT | $\Delta$ SFT | $\Delta$ Chart-RVR |
|---|---|---|---|
| ChartQA | +0.27 | +0.29 | **+0.38** |
| ChartQA (human) | +0.21 | +0.21 | **+0.25** |
| PlotQA | +0.1605 | +0.1642 | +0.1632 |
| EvoChart | +0.16 | +0.14 | **+0.23** |
| ChartQAPro | +0.049 | +0.043 | **+0.058** |

## A.8 FAILURE CASES

Finally, in Figure 8, we report 2 failure cases of Chart-RVR on the OOD dataset EvoChart. The top chart is composed of both line and bar graphs together, making the reasoning process confounded. Note that Chart-RVR correctly identifies the chart type, i.e., line, which is the correct chart to look for, but answers the approximate value 6, which is very close to 5 and can be attributed to a scaling issue. In the next chart, the values of both lines in 2023 are extremely overlapping; hence, all approaches make errors.

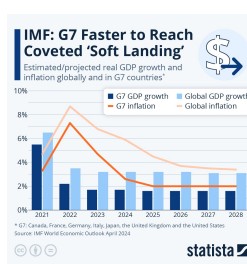

**Query:** ['What is the value of G7 inflation in year 2023?']
**Ground Truth:** ['5']

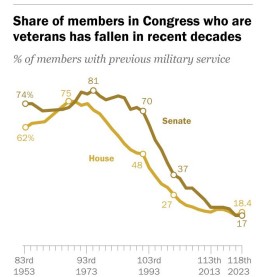

**Query:** ['How many percent of house members have previous military sevice in 2023?']
**Ground Truth:** ['17']

**CoT Rationale:**
<type>\nbar\n</type>
<table>{\ "columns": [ "year", "G7 GDP growth","Global GDP growth",... "rows": ["2021", 5.6, 6.4, 7.1] </table>
<step-1> The chart shows three bars representing G7 inflation for different years. We need to find the value of G7 inflation in 2023.
<step-2> In the year 2023, the bar representing G7 inflation is located at the intersection of the year 2023 and the orange bar
<step-3> The height of the orange bar in 2023 indicates the value of G7 inflation for that year.
<step-4> The value of G7 inflation in 2023 is 3.2%.
<answer>3.2</answer>

**SFT Rationale:**
<type>bar</type>
<table>{'columns': ['Year', 'G7 GDP growth', 'Global GDP growth', 'G7 inflation', 'Global inflation'], 'rows': [['2021..</table>
<step-1>: Identify the relevant data point for G7 inflation in 2023.
<step-2>: Locate the year 2023 in the chart and find the corresponding value for G7 inflation.
<step-3>: Verify that the value for G7 inflation in 2023 is indeed 2.5%.
<step-4>: Confirm that there are no other conflicting values or errors in the data.
<answer>2.5</answer>

**Chart-RVR Rationale:**
<type>Line</type>
<table>{"columns": ["Year", "G7 GDP growth", "Global GDP growth", "G...n"rows": [["2021", 5.8, 6.4, 3.8, 6.5],</table>
<step-1>: Examine the chart, identifying the year 2023. This year corresponds to 2023.
<step-2>: Gather the value associated with G7 inflation for this year. It represents the data for 2023.
<step-3>: Verify the graph, confirming it shows the value associated with each year.
<step-4>: Locate the information for 2023, comparing it to the values for G7 inflation.
<step-5>: Verify the value, which is 6.0% for 2023.
<step-6>: Provide a direct answer to confirm the value of G7 inflation in 2023 is 6.0.
<answer>6.0</answer>

**CoT Rationale:**
<type>line</type>
<table>{"columns": ["decade", "House", "Senate"], "rows": [["83rd", "74%", "75%"], ["93rd", "62%", "81%"],</table>
<step-1>First, I need to identify the relevant data points for the House and Senate in the year 2023. The chart shows that in the 118th decade (which corresponds to 2023), the House has 18.4% of its members who have previous military service, while the Senate has 17%.
<step-2>Next, I should confirm if there is any additional information needed to answer the question. In this case, the data provided directly answers the question without requiring further analysis.
<step-3>Finally, I can conclude that the percentage of House members who have previous military service in 2023 is 18.4%.
<answer>18.4</answer>

**SFT Rationale:**
<type>line</type>
<table>{'columns': ['Year', 'House', 'Senate'], 'rows': [['83rd 1953', '74', '62'], ['93rd 1973', '75', '81'],</table>
<step-1>: Identify the relevant data point for the House in 2023.
<step-2>: From the chart, the value for the House in 2023 is approximately 18.4%.
<step-3>: Verify that there are no other conflicting values or annotations for 2023.
<step-4>: Confirm that the answer matches the provided answer of 18.4.
<step-5>: VERIFY the final answer is correct for no hallucinations.
<answer>18.4</answer>

**Chart-RVR Rationale:**
<type>Line</type>
<table>{"columns": ["Year", "House", "Senate"],"rows": [["83rd 1953", 74, 62], ["93rd 1973", 75, 81],</table>
<step-1>: Examine the chart, identifying the year 2023 for the House members.
<step-2>: Gather the percentage value associated with this year, comparing it to the data for the House.
<step-3>: Verify the information, noting the percentage of 2023 is 18.4%.
<step-4>: Confirm it represents the House members\' percentage of having previous military service in 2023.
<step-5>: Provide a direct answer, verifying the answer is 18.4%.
<answer>18.4</answer>

Figure 8: Example failure case from the EvoChart dataset (OOD) where all Chart-RVR, CoT, and SFT output wrong answers.

## A.9 HUMAN STUDY

Finally, to ascertain the interpretability of the rationales generated by our approach as compared to SFT and CoT, we conduct a human study with 5 graduate-level volunteers. Our study has 5 data samples from the ChartQA and EvoChart datasets, which are **correctly** classified by all 3 approaches - CoT, SFT, and Chart-RVR, and the traces output by these approaches. We do not disclose which method produces what trace, and also shuffle the options. We show the instructions for the study, a sample question, and the final results in Figure 9. As can be observed, most people prefer Chart-RVR responses for 4/5 questions as compared to SFT and CoT, implying a clear preference.

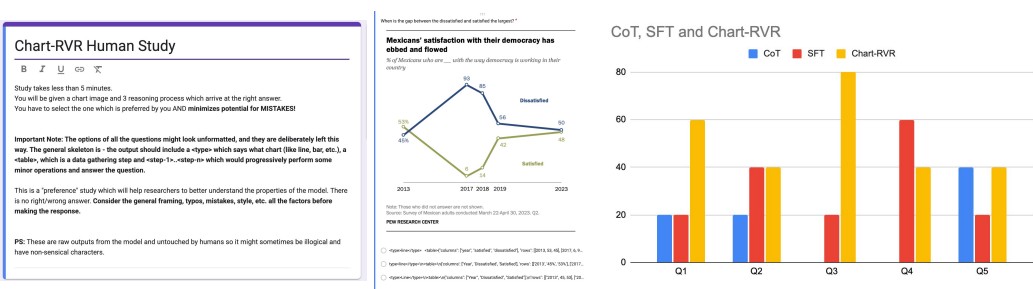

Figure 9: Layout and results of the human study. (LEFT) Instructions to the volunteers, (MIDDLE) Sample Question, and (RIGHT) Percentage of people preferring each of the 3 approaches.

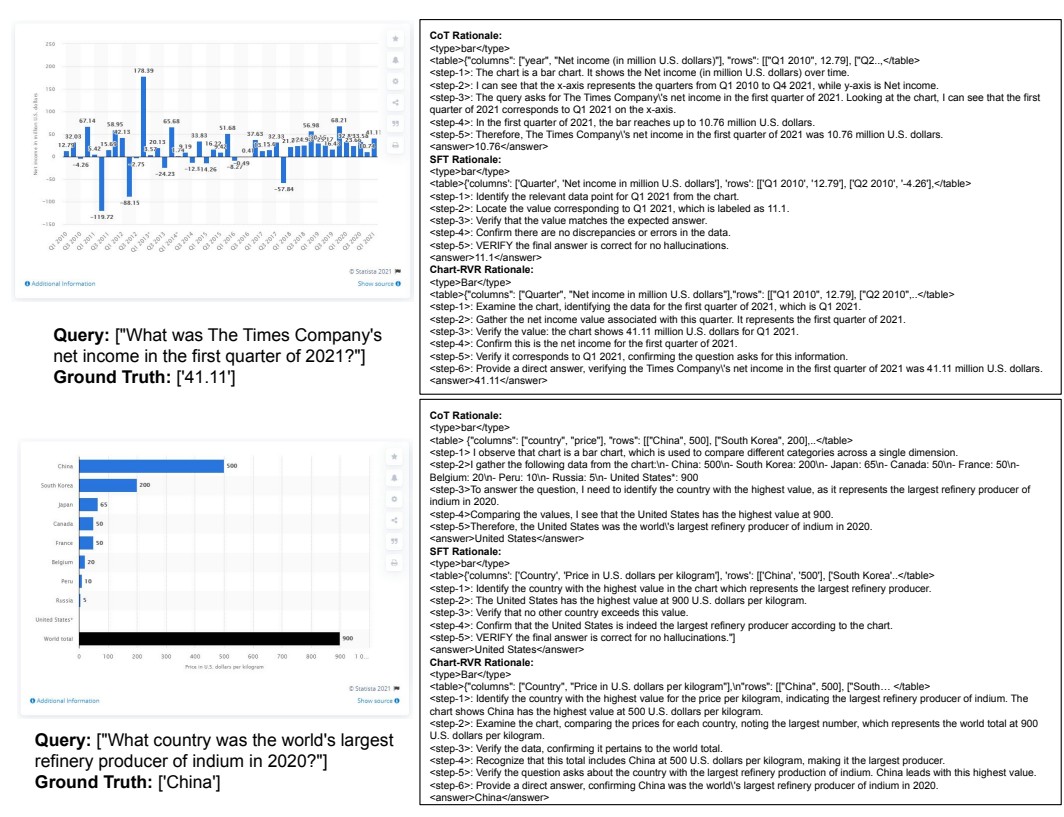

Figure 10: Example failure cases from the ChartQA dataset (ID) where Chart-RVR outperforms CoT and SFT on 2 challenging examples.

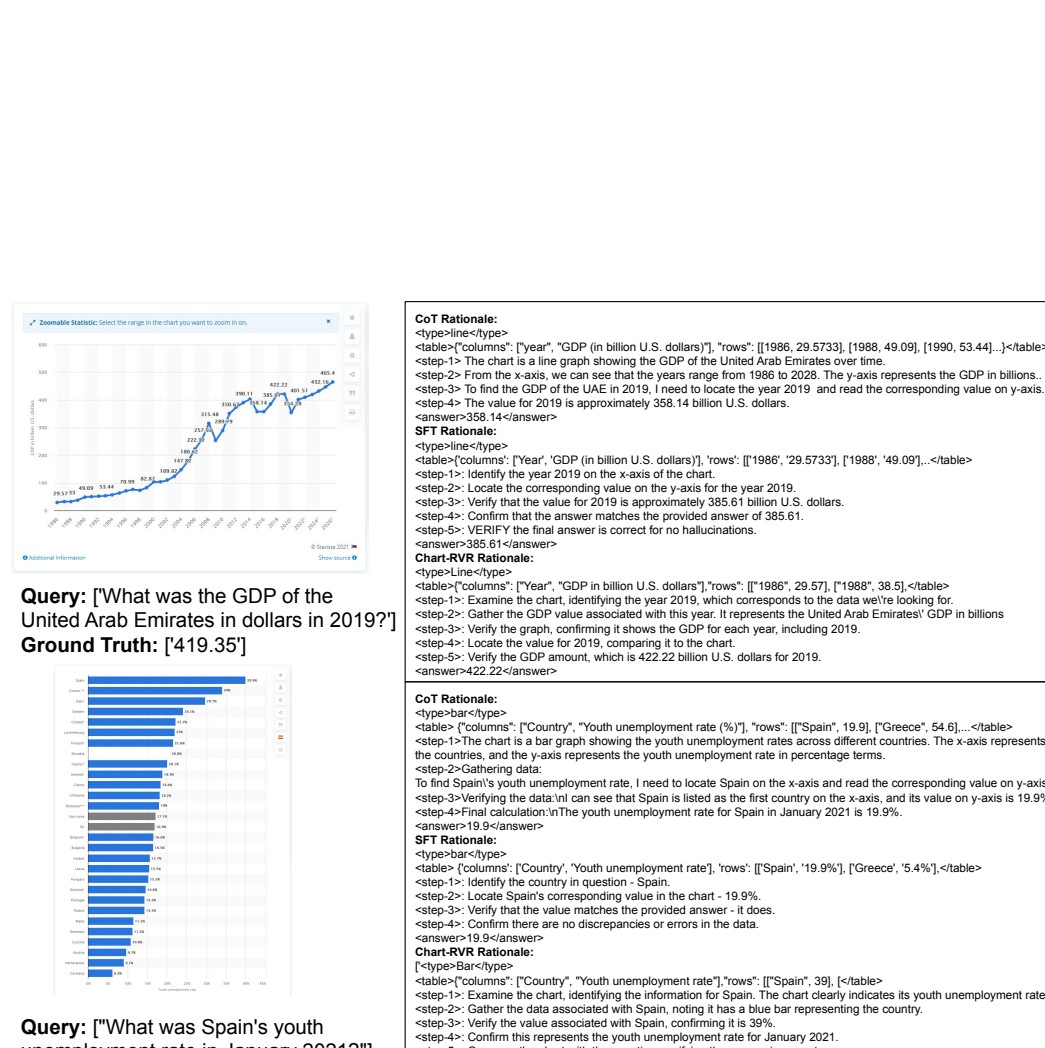

Figure 11: Example failure cases from the ChartQA dataset (ID) where Chart-RVR outperforms CoT and SFT on 2 challenging examples.

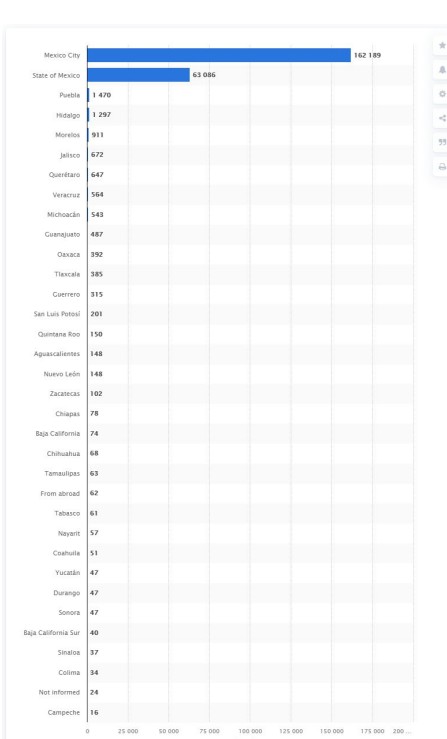

**Query:** ['How many patients came from the neighboring state of Mexico?']
**Ground Truth:** ['63086']

**CoT Rationale:**
<type>bar</type>
<table>
{"columns": ["State", "Number of patients"], "rows": [["Mexico City", 167189], ["State of Mexico",...
</table>
<step-1> The chart is a bar chart showing the number of patients from different states in Mexico.
<step-2> From the chart, I can see that the state of Mexico has the highest number of patients at 63,085.
<step-3> The question asks specifically about patients coming from the neighboring state of Mexico, which is the State of Mexico itself.
<step-4> Therefore, the answer is the number of patients from the State of Mexico, which is 63,085.']
<answer> 63085 </answer>

**SFT Rationale:**
<type>bar</type>
<table>
{\'columns\': [\'State\', \'Number of patients\'],
\'rows\': [[\'Mexico City\', \'67085\'], [\'State of Mexico\', \'470\'], [\'Puebla\', \'297\'],
</table>
<step-1>: Identify the relevant data for the neighboring state of Mexico which is "State of Mexico".
<step-2>: From the chart, the number of patients from the State of Mexico is 470.
<step-3>: Verify that there are no other states listed as being part of the neighboring region.
<step-4>: Confirm that the answer matches the data from the chart.
<step-5>: VERIFY the final answer is correct for no hallucinations.
<answer> 470 </answer>

**GRPO Rationale:**
<type>Bar</type>
<table>
{"columns": ["State", "Number of patients"],
"rows": [["Mexico City", 162189], ["State of Mexico", 63086], ["Puebla", 1470],....
</table>
<step-1>: Identify the information related to the neighboring state of Mexico, which is the State of Mexico. This state has 63,086 patients, indicating it is the neighboring state in question.
<step-2>: Examine the chart, comparing it to the data for Mexico City, the largest state, with 162,189 patients.
<step-3>: Verify the number of patients associated with the State of Mexico, noting it has 63,086 patients.
<step-4>: Confirm this represents the neighboring state of Mexico, as it is the state adjacent to Mexico City.
<step-5>: Verify the question asks for the number of patients coming from this neighboring state. The answer is 63,086.
<step-6>: Provide a final calculation and reasoning, confirming the answer is correct, as it corresponds to the patients from the State of Mexico.
<answer> 63086 </answer>

Figure 12: Example failure case from the ChartQA dataset (ID) where Chart-RVR outperforms CoT and SFT

