# OpenReview forum: "Chart-RVR: Reinforcement Learning with Verifiable Rewards for Explainable Chart Reasoning"
_ICLR.cc/2026/Conference — Submitted to ICLR 2026_

### Official Review · Reviewer_sUe3 · 2025-10-31

**Soundness:** 2
**Presentation:** 2
**Contribution:** 2
**Rating:** 4
**Confidence:** 4

**Summary:**

The paper introduces Chart-RVR, a reinforcement learning framework built on GRPO with verifiable surrogate rewards for chart reasoning.  method combines three components: (1) chart-type prediction, (2) chart-table reconstruction, and (3) a process-conformity reward to enforce structured reasoning. Experiments on six benchmarks show modest accuracy gains over SFT baselines using 3B-parameter LVLMs like Qwen2.5VL. authors claim improved out-of-distribution generalization and more explainable CoT rationales.

**Strengths:**

- The work addresses a recognized limitation of current chart reasoning models i.e. over-reliance on sft and lack of verifiable reasoning.
- modular, verifiable reward components (chart-type, table reconstruction, format, etc.) form a clear and interpretable pipeline that could be useful for future chart-to-text RL research.
- Multiple benchmarks (ChartQA, PlotQA, ChartFC, etc.) are used, providing some breadth of evaluation.
- Despite dense math , paper is logically structured and  states goals, components, and datasets.

**Weaknesses:**

- While the paper positions itself as a "general RL framework for explainable chart reasoning" the technical core is incremental. e.g. reward functions: format, accuracy, type, table and text similarity are largely deterministic existing heuristics, not new learning principles. GRPO has already been used in multiple prior multimodal reinforcement fine-tuning works. The contribution mainly lies in repackaging standard verifiable checks into a chart-specific recipe, without deeper theoretical or algorithmic advancement.
- The empirical study lacks strong evidence that Chart-RVR truly improves generalization or explainability. May be its just me but the CoT explainability metric is bit unconventional and largely depends on a large external LVLM as oracle. some qualitative diversity analysis would have been nice
- Also, worried about ptential biases:  dataset construction (Section 4.1 and appendix A1) relies heavily on Qwen2.5VL-72B-generated rationales as "ground truth" filtered with a few heuristics and minimal human verification. manual filtering step can be better quantified and is currently insufficiently detailed.
- interpretability evaluation is minimal, "explainable Info gain" metric is neither standard nor clearly validated against human judgment.


summary: the motivation and modular reward framework are solid starting points, the idea is promising but the work needs deeper validation, clearer ablations, and stronger writing and presentation before reaching top-tier readiness, I believe.

**Questions:**

- Do authors present ablation or sensitivity results for the numerous reward weights? I may have missed spotting.
- minor: igures showing CoT examples are too small and contain color-coded text that is hard to read. Could you consider upgrading?
- missing references of some relevant papers on visual reasoning and visual RL:
[1] Masry et al. BigCharts-R1: Enhanced Chart Reasoning with Visual Reinforcement Finetuning, https://arxiv.org/abs/2508.09804
[2] Rodriguez et al, BigDocs: An Open Dataset for Training Multimodal Models on Document and Code Tasks. https://arxiv.org/abs/2412.04626
[3] Awal et al. WebMMU: A Benchmark for Multimodal Multilingual Website Understanding and Code Generation https://arxiv.org/abs/2508.16763.

---

> ### Author Response · Authors · 2025-11-19
> **Regarding contributions, generalization and metric choice.**
>
> We appreciate that the reviewer finds our work clear, interpretable and having a wide breadth of evaluation.
>
>
> > on "repackaging"
>
> We would like to re-clarify some core tenets of Chart-RVR:
> Our method goes beyond just adapting GRPO to charts using standard GRPO rewards.
> - Prior multimodal GRPO works primarily target larger models with strong out-of-the-box reasoning and abundant compute. In contrast, Chart-RVR is explicitly designed for the small LVLM regime.
>  - Our surrogate task rewards **improve** factuality by incentivizing high fidelity data gathering in the form of reconstructing the underlying chart type and data table (Table-3b).
>  - Existing multimodal GRPO-based methods for charts (and VQA) typically treat the model as a black-box answer generator and assume accuracy/format/length rewards are sufficient to ellicit good reasoning. Our process-conformity reward alleviates this by isolating and separately controlling the "data-gathering" and "reasoning" steps as demonstrated qualitatively in Figure-1 and Appendix. As can be seen, focussing on only one aspect of reasoning at each step better disentangles traces, improving the explainability and correctness.
>
> > The empirical study lacks strong evidence that Chart-RVR truly improves generalization
>
> We have meticulously ensured our OOD data benchmarks are as strong as possible. In Chart-RVR, we have evaluated OOD data against 3 real life complex OOD datasets - EvoChart, ChartQAPro, ChartBench and CharXiv (please see qu1R response) - making our OOD data evaluation robust.
>
>
> > CoT explainability metric .... unconventional
>
> The explainability metric leverages the strong reasoning capabilities of the Oracle model (to judge the reasoning), which assigns higher token log-likelihood probability scores if the given rationales appended to the query increase the probability of correct answer token. We follow a long lineage of delta-based faithfulness literature to design the metric, first proposed in [1] and expanded in [2,3] to model our explainability metric.  We also utilize a different family of Oracle model - LLaVA-Next-72B in Appendix (Table-6) with identical conclusions across all datasets (Chart-RVR improves wrt SFT on OOD data).
>
>
>
>
> > on dataset construction
>
> To clarify, we add more details on the dataset validation:
> 1. For pass 1, the prompt structure shown in Figure 2 is used to generate the chain-of-thought data.
> 2. For pass 2, all samples with overtly short/long and incorrect reasoning traces are filtered (i.e. trace length less than 3 or more than 8). To validate if a trace is wrong, we algorithmically check if the last CoT line trace contains the correct answer. 358 samples in total are filtered out in this step from a total of 15000.
> 3. For pass 3, three doctoral-level researchers are employed to validate the reasoning traces of a randomly sampled data subset (1000 samples). Each researcher filtered less than 6 samples (<1%) in total with a 100% agreement between them.
>
> We have reported these details in the updated PDF.
>
> > on explainability evaluation
>
> In addition to reporting the numbers on the explainability metric as detailed in Section 4.6 on Qwen-72B (Table-4) and LLaVA-Next-72B (Table-6 Appendix), we also conduct a human study measuring user preferences to CoT, SFT and Chart-RVR in Figure-8 (Appendix). Note that our study is conducted on samples where all three approaches give the correct answer, removing the confounding factor of "answer correctness", and only evaluates which reasoning trace is the most explainable to humans.
>
>
>
> > ablation for reward weights
>
> In our reward design, tunable *reward weights* are $\lambda_1$, $\lambda_2$ and $\alpha$ and are discussed in detail in Appendix lines 759-767 for Qwen Models. We observe that due to smooth normalization, minor changes to each hyperparameter does not affect performance significantly. The hyperparameters are needed as our method is generalizable to diverse model families having different "styles" of reasoning and can require scaling of rewards depending on underlying RL library (like TRL, verl, etc.). We will include a broader discussion around Gemma and InternVL3.5 family of models in the final version, as each RL ablation run requires extensive compute.
>
>
> > text that is hard to read and missing references
>
> We acknowledge the concern and have increased the font size of figures. We will also cite the relevant papers mentioned. We would also like to point reviewer to our general response regarding BigCharts-R1.
>
> [1] ERASER: A benchmark to evaluate rationalized NLP models, ACL 2020 \
> [2] Measuring Chain of Thought Faithfulness by Unlearning Reasoning Steps, EMNLP 2025 \
> [3] Making Reasoning Matter:
> Measuring and Improving Faithfulness of Chain-of-Thought Reasoning, EMNLP 2024

---

> ### Author Response · Authors · 2025-11-24
> **We are happy to answer any further questions**
>
> Dear Reviewer sUe3, As the author-reviewer discussion period is winding down, we would like to thank you for the time and effort you have invested in reviewing our submission.
>
> If you have any unresolved concerns, please let us know, and we are happy to address them.
>
> Thank you.

---

> ### Author Response · Authors · 2025-11-27
> **Additional Human validation study to verify Oracle annotations**
>
> Dear Reviewer sUe3,
>
> Following your suggestion (W3), we manually validated 100 randomly sampled oracle-annotated charts from EvoChart and ChartQAPro datasets (note no ground truth table annotations are available for either).
>
> We categorize each data sample into 2 categories -
> 1. "Needs Approximation?" - which entails guessing the correct values from the chart axes rather than labelled counterparts
> 2. "Factoid/Direct" - which entails looking at axes marks to determine the correct values.
>
> Below, we report the Kendall's Tau: the inter-annotator agreement score b/w human (annotator-1) and Oracle model (annotator-2).
>
> | Category           | #Samples  | Cohen’s Kappa |  |
> |--------------------|----------|-----------------|-----------------------|
> | Needs Approximation | 48  |  0.91           |
> | Factoid/Direct     |  152             |  0.98               |
> | **Overall**        | 200               | 0.97                 |
>
>
> Together with our 3-part dataset sanity check, this suggests that the Qwen2.5-VL-72B annotations are high-quality approximations for surrogate rewards/metrics.
>
> We have updated the PDF with this table and a qualitative failure case as well.
> We thank you again for the review and are happy to answer any questions.

---

### Official Review · Reviewer_qu1R · 2025-10-31

**Soundness:** 3
**Presentation:** 2
**Contribution:** 2
**Rating:** 2
**Confidence:** 4

**Summary:**

This paper proposes Chart‑RVR, a reinforcement‑learning fine‑tuning framework for large vision‑language models (LVLMs) for chart-reasoning. In this work, they explore GRPO with
A set of verifiable rewards: (surrogate tasks): chart-type classification, underlying chart-table reconstruction and process-conformity reward that measures step-by-step reasoning aligned with ground truth rationales. In this work the authors fine-tune and evaluate 3B (Qwen 2.5VL) models across six chart QA test datasets that constsis of both in-domain and OOD. The overall results show modest gains on in-domain (1 - 2%) and larger gain (3.5 - 7% on OOD test sets). They show that the proposed framework generalizes to multiple LVLM architectures ( Gemma3 and InternVL3.5-4B ).

**Strengths:**

1. GRPO with multiple verifiable surrogate tasks and process conformity reward.
2. Ablation on how the surrogate tasks help with over all performance.
3. Benchmarking on multiple datasets both in and out of distribution and showing that the proposed framework improves well on OOD.
4. Demonstrating this framework is generalizable across various architectures.
5. Human study that shows humans prefer chart-rvr reasoning over others (on a very small sample though)

**Weaknesses:**

1. All the models studied here are around 3B. Would it generalise to bigger models? Would SFT alone be sufficient for bigger models (if there is a way of getting larger, cleaner data). See BigCharts-R1 paper on how to get such data.

2. A missing citation to very relevant work BigCharts-R1 who effectively propose an identical framework. Instead of using surrogate tasks in RL they use that to generate/synthesise large finet-tuning data and complement with RL fine-tuning on top of it.

3. Missing numbers on CharXiv dataset that specifically tests the questions that truly require reasoning.

BIGCHARTS-R1: Enhanced Chart Reasoning with Visual Reinforcement Finetuning: https://arxiv.org/pdf/2508.09804

**Questions:**

1. Since ChartQA (not really sure if you have used same PlotQA subset or all of it) is the only common dataset between BigCharts-R1 and this work, looks like SFT on high quality data does as well as your RL framework. Any explanation on this?

2. Does higher process‑conformity scores correlate with higher answer accuracy? Curious if this is the case and if there are any patterns observed otherwise?


3. Would like to see a large human study on data sampled from multiple datasets or different level of visual/complex reasoning and if chart-rvr would still be preferred over other models wich larger data sample?

4. How did you choose the length thresholds for length reward?

5. For the table reconstruction (looks like model would completely penalize if the values predicted in cell are not exact but close enough). Could be a real issue if you have data where model needs to estimate from the visuals or infographics of the plot. If not a lot of such data is present it is a issue of the model to generalize to such data.

6. Would be good to see that this would generalize across different model sizes (just like different model families). Perhaps smaller models if there is constraints on working on larger models.

---

> ### Author Response · Authors · 2025-11-19
> **Comparison to BigCharts and updated CharXiv results**
>
> We thank the reviewer for pointing us to the relevant work BigCharts-R1. As BigCharts-R1 appeared Aug 13, 2025 (arXiv) and Aug 25, 2025 (COLM OpenReview), after the ICLR 2026 contemporary work cutoff (July 24) https://iclr.cc/Conferences/2026/ReviewerGuide, which states - "That means, since our full paper deadline is September 24, if a paper was published (i.e., at a peer-reviewed venue) on or after July 24, 2025, authors are not required to compare their own work to that paper". Nevertheless, we compare and contrast our approach - please refer to our consolidated response.
>
> We also appreciate that the reviewer finds our work with ample ablations, OOD benchmark, and generalization to other architectures.
>
> > on generalization to bigger models
>
> Due to extensive computational requirements, we report the results on 1 epoch (instead of 4) on "Qwen2.5-VL-7B-Instruct" due to resource constraints. We report the results on ChartQA (ID) and our OOD benchmarks. We observe identical trends as seen in 3 billion models, wherein Chart-RVR improves OOD generalization.
>
> | Method   | ChartQA | EvoChart |ChartQAPro  | ChartBench |
> |----------|--------:|---------:|-----------:|-----------:|
> | CoT      |  76.12  |  41.84   | 26.12      | 69.66      |
> | SFT      |  86.72  |  55.6    | 26.64      | 67.36           |
> | Chart-RVR| 86.6    | 58.4     | 37.36      |  70.48          |
>
>
>
> > on more data
>
> Please refer to our general response on "Core contributions"
>
> > citation to BigCharts-R1
>
> Please refer to our general response on "Comparision to BigCharts-R1".
>
> > Charxiv numbers
>
> Please find below numbers on the Charxiv Reasoning subset on our Qwen2.5VL-3B models. Similar to the trends reported, Chart-RVR outperforms on the reasoning subset.
>
> | Method   | CharXiv (reasoning)
> |----------|---------:|
> | CoT      |   26.2   |
> | SFT      |   29.6   |
> | Chart-RVR|   34.3   |
>
>
> > Q1. looks like SFT on high quality data does as well as your RL framework.
>
> Please refer to our general response on "Core contributions"
>
> > Q2. .. higher process‑conformity scores correlate with higher answer accuracy
>
> Yes, process conformity is a critical component in improving performance, as demonstrated in Tables-1 and 5. For models with no process conformity, i.e. A+F+L+Tasks, the performance on OOD improves modestly over standard SFT/GRPO, while process conformity (Chart-RVR) pushes the numbers to SOTA values. We also refer reviewer to plots in Figure-4 (Appendix), which show the dynamics of reward values (including process conformity) during fine-tuning.
>
> > Q3. on human study
>
> A validation study with 5 highly qualified volunteers is conducted and results of which are reported in Appendix A.8. We obseve our volunteers preferring Chart-RVR reasoning traces over CoT and SFT across all questions. We plan to conduct a large scale study in the future.
>
>
> > Q4. Length thresholds
>
> The thresholds are chosen as lower and higher bounds of generated reasoning data from ground-truth reasoning dataset's trace lengths with a 50 token slack at both ends. We observe that overtly long traces tend to reward hack by repeating reasoning facts, hence an upper limit is enforced. We have provided a discussion in Section A.3.
>
> > Q5. On table reconstruction
>
> Please refer to our response on dataset construction. We assume that the "correct" table interpretation is performed by the Oracle model utilized to construct the dataset.
>
> > Q6. on model sizes
>
> Our methodology is tested out on the biggest models, which can be deployed on the edge. The biggest challenge with models below 3 billion parameters is the almost non-existent CoT performance out-of-box (for e.g. SmolVLM, which gives only a 12.4% RA using CoT on ChartQA), which are trained to predict final output and do not provide valid CoT traces.

---

> ### Author Response · Authors · 2025-11-24
> **We are happy to answer any further questions**
>
> Dear Reviewer qu1R,
> As the author-reviewer discussion period is winding down, we would like to thank you for the time and effort you have invested in reviewing our submission.
>
> If you have any unresolved concerns, please let us know, and we are happy to address them.
>
> Thank you.

---

### Official Review · Reviewer_Sdaj · 2025-10-31

**Soundness:** 2
**Presentation:** 3
**Contribution:** 3
**Rating:** 6
**Confidence:** 3

**Summary:**

This paper presents Chart-RVR, a reinforcement learning framework for improving the robustness and interpretability of chart reasoning in LVLMs. Chart-RVR aims to address the OOD generalization and unreliable CoT reasoning issues in LVLMs. It combines GRPO with automatically verifiable rewards, introducing three reward types to ensures models identify the chart type, measures how accurately the model reconstructs the underlying data table, and enforces stylistic and structural consistency in reasoning steps. Extensive experiments across six chart benchmarks show that Chart-RVR-trained models outperform SFT and domain-specific baselines on OOD datasets.

**Strengths:**

This paper introduces a verifiable reward structure integrated into GRPO, enabling stable and interpretable reinforcement fine-tuning for LVLMs. Unlike SFT, Chart-RVR directly optimizes verifiable task outcomes to improve robustness and reasoning interpretability.

Chart-RVR achieves outperforming accuracy across six benchmarks, with the significant accuracy gains under OOD settings. The Process-Conformity Reward effectively enforces reasoning alignment with ground truth, yielding more coherent traces. The proposed explainable information gain further demonstrates that Chart-RVR rationales increase model confidence and interpretability on harder datasets.

**Weaknesses:**

A more detailed discussion about the interaction dynamics between multiple verifiable rewards (e.g., balancing \lambda_1 and \lambda_2 in Eq. 6) are recommended. The training data is generated using Qwen2.5VL-72B. This raises concerns about data bias, as the quality are not verified by human at scale. What will the accuracy change if the data is constructed using different LVLMs?

All benchmarks are chart-based. I wonder if the proposed approach can be applied to non-chart-based reasoning tasks. There is no experiments assess whether the framework generalizes to other structured visual reasoning tasks. So the generality of RVR beyond charts thus remains unclear.

The \delta logP improvements in Table 4 show clear OOD benefits but relatively small or negative gains in ID settings (e.g., ChartQA).

**Questions:**

How sensitive is performance to the weighting of \lambda_1 and \lambda_2 in Eq. 6?

Given that the CoT datasets are generated by a relatively large LVLM (Qwen2.5VL-72B), will the results drop with a smaller generator?

Could the RVR framework be applied to other domains like general visual-language reasoning tasks? What modifications would be required?

Does enforcing strict process conformity reduce flexibility or creativity in reasoning, e.g., alternative but valid reasoning paths?

---

> ### Author Response · Authors · 2025-11-19
> **Details on hyperparameters, dataset construction and generalization beyond charts**
>
> We are extremely thankful to the reviewer for providing a very high quality review.
>
> > on hyperparameters:
>
> In our reward design, tunable *reward weights* are $\lambda_1$, $\lambda_2$ and $\alpha$ and are discussed in detail in Appendix lines 759-767 for Qwen Models. We observe that due to smooth normalization, minor changes to each hyperparameter does not affect performance significantly. The hyperparameters are needed as our method is generalizable to diverse model families having different "styles" of reasoning. We will include a broader discussion around Gemma and InternVL3.5 family of models in the final version.
>
> > on dataset
>
> We acknowledge the concern. We also report the dataset construction procedure here for completeness (will be incorporated in the final version):
>
> To clarify, we here add more details on the dataset validation:
> 1. For pass 1, the prompt structure shown in Figure-2 is used to generate the chain-of-thought data.
> 2. For pass 2, all samples with overtly short/long and incorrect reasoning traces are filtered (i.e. trace length less than 3 or more than 8 based on manual inspection and outlier rejection of overtly short and long, confusing reasoning rationales). To validate if a trace is wrong, we algorithmicaly check if the last CoT line trace contains the correct answer. A total of 358 samples are filtered out in this step out of 15000.
> 3. For pass 3, three doctoral level researchers are employed to validate the reasoning traces of a randomly sampled data subset (1000 samples). Each researcher filtered less than 6 samples (<1%) in total with a 100% agreement between them.
>
>
>
> > on only chart based benchmarks
>
> The reviewer raises an interesting question. Usually GRPO on smaller LVLMs is challenging as the diversity of rollouts generated using the policy model is low - unable to capture sparse rewards. To alleviate this, some methods utilize SFT as "bootstrapping" [1] which can overfit to surface patterns and limit OOD gains. Our work uses the Process Conformity Reward as a proxy for generating dense rewards, easier for smaller LVLMs to capture. Multiple newer lines of research [2] have been utilzing inter-step reasoning agreement and [3] on policy distillation i.e. matching tokens in each reasoning step to alleviate the sparse reward problem. Hence, our method is a general approach to fine-tune smaller LVLMs on benchamrks in addition to charts. To verify our hypothesis, we run a SFT, GRPO and Chart-RVR algorithm (w/o chart-specific rewards, only process conformity - labeled as "Ours") on a small subset of the CLEVR dataset (500 samples for train/test). We observe our approach outperforms SFT and GRPO, implying generalization beyond charts (Note: CoT data is generated similar to Chart-RVR).
>
> | Method   | Accuracy |
> |----------|---------:|
> | Base     |     53.8 |
> | SFT      |     87.6 |
> | GRPO     |     86.4 |
> | Ours     | **89.8** |
>
>
>
>
> > on small improvement in ID samples
>
> We appreciate the eagle-eyed nature of the reviewer. ChartQA, PlotQA and ChartFC are heavily benchmarked datasets and part of pre-training recipes of multiple frontier open-source models [Refer page 12 in https://arxiv.org/abs/2412.05271], exhibiting high stylistic overlap with pretraining and allow shortcut solutions. This reduces headroom for gains on ID data.
>
>
> > Q1 ...hyperparameters
>
> Please refer to our response on hyperparameters.
>
> > Q2 ...smaller data generators
>
> As larger models posess larger world knowledge and reasoning skills, we utilize the largest model (we could inference on our resources) for our setup. A smaller model can theoretically be utilized, but can suffer from outputting sub-par reasoning traces.
>
> > Q3. ...RVR framework be applied to other domains
>
> Yes, and we believe that is one of the critical contributions of our methodology. We utilize RVR for CLEVR (counting task) with gains over SFT and GRPO as shown above.
>
> > Q4. ... reduce flexibility or creativity
>
> For a structured problem like Chart reasoning (or counting), the reasoning pathways are not very diverse due to a deterministic way to reach the final answer (hence only reporting Pass@1 accuracy). We have shown some examples of how the reasoning traces change in Figure 1.
>
>
>
> [1a] Visual-RFT: Visual Reinforcement Fine-Tuning, ICCV 2025 \
> [1b] https://arxiv.org/abs/2504.07615 \
> [2] Supervised Reinforcement Learning: From Expert Trajectories to Step-wise Reasoning, Arxiv 2025 \
> [3] https://thinkingmachines.ai/blog/on-policy-distillation/

---

### Official Review · Reviewer_N8w4 · 2025-11-01

**Soundness:** 3
**Presentation:** 3
**Contribution:** 3
**Rating:** 6
**Confidence:** 4

**Summary:**

This paper introduced Chart-RVR an RL training framework to improve the reasoning in chart understanding models and the explainability of the reasoning steps. The authors utilize the GRPO RL algorithm and propose three rewards: chart type prediction, chart table reconstruction, and process conformity. The process conformity evaluates each generated reasoning step against a gold step from an oracle model (Qwen2.5-VL-72B) using text embedding similarity. The authors conducted extensive experiments showing the superiority of their approach compared to SFT and other simpler RL approaches that only rely on the final answer accuracy and format. The authors also evaluated their model on a dverse set of chart reasoning tasks such as Chart QA, chart fact checking, chart type classification and chart table reconstruction.

**Strengths:**

* Extensive evaluation on diverse tasks and benchmarks, including chart question answering, fact checking, chart classification, and chart table reconstruction. The Chart-RVR model achieves strong results on most benchmarks proving the proposed approach effectiveness.
* The authors also show the generalization of their approach across different LLM architectures such as InternVL and Gemma. This is quite important in my opinion because most recent RL papers only QwenVL and their approaches do not generalize to other pretrained models.
* The authors also analyzed the interpretability and explainability of the generated rationale by their model showing that their RL approach achieves better explainability than the SFT approach.

**Weaknesses:**

* The authors have not provided any ablation studies to show the impact & importance of each reward on the model performance. I believe the proposed rewards are overengineered, especially the process conformity reward. It would be helpful to support these claims and design choices by running some ablation studies by removing one reward at a time and showing the performance.


* There are limited details about the dataset used for training the model. The authors should analyze the dataset and provide some insights (e..g, quality check).

**Questions:**

In Table 2b, the authors report the performance of the model on surrogate tasks like Chart type prediction and Table reconstruction. However, some of the listed datasets such as ChartQAPro do not provide any ground truth chart types or data tables. I am wondering how did the authors evaluate the output of their model on such unlabeled benchmarks.

---

> ### Author Response · Authors · 2025-11-19
> **Details on reward ablations and data construction**
>
> We appreciate that the reviewer finds our work extensive, generalizable across model families and improving explainability.
>
>
> > ablation studies to show the impact & importance of each reward
>
> We have reported the numbers around reward ablations in Table-2 where row **(A+F+L)** represents standard **GRPO**, **(A+F+L+Tasks)** represents GRPO with **surrogate rewards** and **Chart-RVR** represents **GRPO with surrogate task rewards and the process conformity rewards (Ours)**.  Additionally, we have also reported the impact of each reward on Gemma3 and InternVL3.5 family in Table-5.
> We restate ablation results on OOD benchmarks here:
>
> | Method   | EvoChart | ChartQAPro | ChartBench |
> |----------|---------:|---------:|---------:|
> | CoT      |   29.6   | 15.80 | 51.16
> | SFT      |   46.08   |  23.56 | 64.64
> | GRPO     |   38.88   |  17.55 | 48.1
> | GRPO+Tasks|   51.68   |  27.66 |  65.28
> | GRPO+Tasks+Process (Chart-RVR) | **53.36** | **28.38** | **68.32**
>
>
> > On Process Conformity
>
> Process Conformity is designed to incentivize factuality and limit the amount of mistakes made by the model during reasoning. Please refer to a visual example in Figure-1a. The SFT model (and CoT) tries to gather 3 things at once - the green line's location, what the green line represents and what is the value of the green line. In doing three tasks together, it makes a mistake and outputs the wrong answer. Our Process Conformity Rewards prevent this behavior and disentangle each step of the reasoning - hereby incentivizing model to perform one step at a time, thus improving reasoning.
>
> >There are limited details about the dataset used for training the model
>
> We acknowledge the concern. We also report the dataset construction procedure here for completeness (will be incorporated in the final version). We utilize dataset construction methods for RL based finetuning. To clarify, we add more details on the dataset validation:
> 1. For pass 1, the prompt structure shown in Figure-2 is used to generate the chain-of-thought data.
> 2. For pass 2, all samples with overtly short/long and incorrect reasoning traces are filtered (i.e. trace length less than 3 or more than 8 based on manual inspection and outlier rejection). To validate if a trace is wrong, we algorithmically check if the last CoT line trace contains the correct answer. A total of 358 samples are filtered out in this step out of 15000.
> 3. For pass 3, three doctoral level researchers are employed to validate the reasoning traces of a randomly sampled data subset (1000 samples). Each researcher filtered less than 6 samples (<1%) in total with a 100% agreement between them.
>
>
> > Q: on unlabeled benchmarks
>
> For datasets without table ground truths (all except ChartQA), we annotate each evaluation dataset's test set using the aforementioned procedure for underlying table and chart type annotations using the same Oracle model - Qwen-2.5VL-72B.

---

> > ### Comment · Reviewer_N8w4 · 2025-11-26
> >
> > Thank you for your response which clarified some of my concerns.
> >
> > I believe using Qwen-2.5VL-72B for annotating the data table is not enough. Some papers such as ChartQAPro show quite limited performance of these models in understanding such complex charts. Hence, I believe human annotations/revisions are necessary, but I keep my current positive evaluation of your work.

---

> ### Author Response · Authors · 2025-11-27
> **Thank you for your support and feedback**
>
> Thank you very much for your thoughtful follow-up and for maintaining your positive evaluation of our work. We truly appreciate the time and care you have devoted to reviewing it.
>
> We fully agree that Qwen2.5-VL-72B–based annotations are not perfect "gold" and therefore use them only as high-capacity teacher signals for surrogate tasks (and rewards), while all core QA evaluations rely on the original human-curated labels.
>
> Our 3-part *manual* dataset sanity check, as described in the rebuttal, provides evidence that these oracle-based annotations are high-quality approximations rather than assumed "perfect gold" data, and in the final version (and dataset release), we will explicitly highlight this limitation and expand human sanity checks to further reassure readers.

---

> > ### Author Response · Authors · 2025-11-27
> > **Additional Human validation study to verify Oracle annotations**
> >
> > Thank you again for raising this important point. Following your suggestion, we manually validated 100 randomly sampled oracle-annotated charts from EvoChart and ChartQAPro datasets (note no ground truth table annotations are available for either).
> >
> > We categorize each data sample into 2 categories -
> > 1. "Needs Approximation?" - which entails guessing the correct values from the chart axes rather than labelled counterparts
> > 2. "Factoid/Direct" - which entails looking at axes marks to determine the correct values.
> >
> > Below, we report the Kendall's Tau: the inter-annotator agreement score b/w human (annotator-1) and Oracle model (annotator-2).
> >
> > | Category           | #Samples  | Cohen’s Kappa |  |
> > |--------------------|----------|-----------------|-----------------------|
> > | Needs Approximation | 48  |  0.91           |
> > | Factoid/Direct     |  152             |  0.98               |
> > | **Overall**        | 200               | 0.97                 |
> >
> >
> > Together with our 3-part dataset sanity check, this suggests that the Qwen2.5-VL-72B annotations are high-quality approximations for surrogate rewards/metrics.
> >
> > We have updated the PDF with this table and some failure cases as well.

---

### Author Response · Authors · 2025-11-19
**Clarifications regarding core contributions and comparision to BigCharts-R1**

**Clarifications regarding core contributions**

Chart-RVR is a methodological recipe for fine-tuning small LVLMs with verifiable rewards.
1. Our motivation is to provide a strong and generalizable fine-tuning methodology that utilizes commonly used ,well-benchmarked/validated datasets (e.g. ChartQA, ChartFC, PlotQA) for LVLM fine-tuning.
2.  Our approach achieves benchmark results on both ID and OOD real-life datasets (average +2-3% on ID +5% on OOD). We have demonstrated this behavior by both - randomly sampling data from the training sets (Table-3a) or data complexity-aware sampling (Table-2: "Hard" suffix).
3. It is architecture agnostic and validated on Qwen-2.5-VL, Gemma-3, and InternVL-3.5.
4. Our approach produces more explainable reasoning traces validated bya  large Oracle model and a human study.

Unlike other chart-reasoning works like [1], we do not curate a new large dataset; instead utilize existing datasets for fine-tuning.

**Salient differences with respect to BigCharts-R1**

We sincerely thank Reviewers qu1R and Sue3 for bringing BigCharts-R1 [1] to our attention. BigCharts-R1 appeared Aug 13, 2025 (arXiv) and Aug 25, 2025 (COLM OpenReview), after the ICLR 2026 contemporary work cutoff (July 24) as stated here: https://iclr.cc/Conferences/2026/ReviewerGuide - "That means, since our full paper deadline is September 24, if a paper was published (i.e., at a peer-reviewed venue) on or after July 24, 2025, authors are not required to compare their own work to that paper".

Nevertheless, upon reading [1] in detail, we believe both works share the high-level goal of improving chart reasoning with RL, however, the concrete contributions and technical focus differ. We detail the differences below:

1. *Divergence in contributions*: [1] proposes BigCharts dataset, which is a completely new dataset composed of training data from existing datasets - Figure QA, DVQA, PlotQA and ArxivQA, along with **chart data collected using Google Search and Common Crawl**. Our work on the other hand focuses on *proposing a methodology* to train (off-the-shelf) smaller LVLMs on existing data from ChartQA, ChartFC and PlotQA with only data annotation steps using a large Oracle model. (Reference Section 3.1.1 in [1])
2. *Reward Design*: Unlike [1], our method not only utilizes standard GRPO rewards (accuracy, length, format as is reported in [1]) but also incorporates factuality specific rewards meticulously designed for chart reasoning - chart type, table reconstruction and process conformity. Particularly, process conformity helps in fine-tuning smaller LVLMs without a SFT phase.
3. *OOD evaluation*: Finally, our work differs wrt [1] on significant OOD testing scenarios. [1] utilizes OOD benchmarks as - FigureQA-subset, DVQA-subset and PlotQA-subset (detailed in Section 5.4). All 3 of these datasets are **synthetic** in nature and do not capture complex and diverse real-life charts. In Chart-RVR, we have evaluated OOD data against 3 real life complex OOD datasets - EvoChart, ChartQAPro and ChartBench - making our OOD data evaluation analysis much more robust. We also report results on CharXiv below.
4. Finally, Chart-RVR, provides analysis around the explainability of the reasoning traces using Oracle and human studies.

We view BigCharts-R1 as a complementary work - focusing on scaling data curation for large models, whereas Chart-RVR focuses on verifiable-reward RL recipes for small LVLMs and explainability. We have cited, compared and contrasted our work with BigCharts-R1 in the updated PDF.

[1] BigCharts-R1: Enhanced Chart Reasoning with Visual Reinforcement Finetuning, COLM 2025

---

### Meta-Review · Area_Chair_jx5m · 2026-01-04

**Summary:**

1. Two reviewers (Reviewer qu1R and sUe3) think the paper's novelty is weakened by a contemporary work, BigCharts-R1, published in COLM 2025.
2. The authors used a powerful "Oracle" model (Qwen2.5VL-72B) to generate ground-truth rationales and annotate datasets, which raised questions about potential data bias and the lack of large-scale human verification (Reviewer Sdaj, sUe3).
3. Reviewer sUe3 argued that the technical contribution was incremental, viewing it as a variant of existing heuristics (GRPO, verifiable rewards) for a specific recipe rather than a fundamental algorithmic advancement.
4. The evaluation is focused on chart-based tasks and smaller (3B) models, leading to questions about the framework's generalizability to general reasoning domains (Reviewer Sdaj) and larger model sizes (Reviewer qu1R).
5. This paper lacks more detailed ablation and sensitivity analyses for the different reward components and their weights (N8w4, Sdaj, sUe3).

**Reviewer Concerns:**

1. The authors point out that BigCharts-R1 was published after the ICLR contemporary work cutoff date, meaning a comparison was not required. They also provide a detailed qualitative comparison, add the citation, and explain that an empirical comparison was impossible as the BigCharts-R1 dataset was not released.
2. The authors conduct a manual validation study on 200 oracle-annotated samples, reporting a very high inter-annotator agreement. However,  the human study was small and measured user preference rather than a direct, objective measure of reasoning faithfulness.
3. The authors add new experiments on the Qwen2.5-VL-7B-Instruct model, showing that the performance trends hold.
4. While the authors are commended for running a new experiment on the CLEVR dataset, this single, small-scale experiment is not sufficient to fully address concerns about the framework's generalization (Reviewer Sdaj). I suggest that authors consider providing extra results on other datasets, such as general visual question answering, math reasoning, and document understanding.
5. A full sensitivity analysis for reward weights was not provided, though the authors did provide some discussion. This is a minor weakness.

**Reviewer Scores:**

The reviewer N8w4 replied that he/she will keep the positive rating. The remaining three reviewers did not provide their feedback. The primary concern for this paper is the incremental nature of the technical contribution. The proposed framework is essentially a well-executed "recipe" that combines existing, off-the-shelf components: the GRPO algorithm and a set of verifiable rewards (e.g., accuracy, format checks, table reconstruction), which have been studied in prior work. While applying these to the chart reasoning task is useful, the work does not introduce a new learning principle or a new insight. Besides,  the CoT explainability metric is a bit unconventional and largely depends on a large external LVLM as an oracle. Although the authors provide a human validation study, the 200 samples are small and measure user preference rather than a direct, objective measure of reasoning correctness and faithfulness.
The limited scope and insufficient validation of its core claims on explainability prevent it from meeting the bar for novelty and impact expected at ICLR.

---

### Decision · Program_Chairs · 2026-01-26

Reject